# A revised dry deposition scheme for land-atmosphere exchange of trace gases in ECHAM/MESSy v2.54

Tamara Emmerichs[1], Astrid Kerkweg[1], Huug Ouwersloot[2], Silvano Fares[3], Ivan Mammarella[4], and Domenico Taraborrelli[1]

[1]Institute of Energy and Climate Research 8, Troposphere, Forschungszentrum Jülich, Jülich, Germany
[2]Max Planck Institute for Chemistry, Mainz, Germany
[3]National Research Council, Institute of Bioeconomy, Rome, Italy
[4]Institute for Atmospheric and Earth System Research / Physics, Faculty of Science, University of Helsinki, Finland

**Correspondence:** Domenico Taraborrelli (d.taraborrelli@fz-juelich.de)

**Abstract.** Dry deposition to vegetation is a major sink of ground-level ozone and is responsible for about 20 % of the total tropospheric ozone loss. Its parametrisation in atmospheric chemistry models represent a significant source of uncertainty for the global tropospheric ozone budget and might account for the mismatch with observations. The model used in this study, the Modular Earth Submodel System (MESSy2) linked to ECHAM5 as an atmospheric circulation model (EMAC), is no exception. Like many global models, EMAC employs a "resistance in series" scheme with the major surface deposition via plant stomata which is hardly sensitive to meteorology, depending only on solar radiation. Unlike many global models, however, EMAC uses a simplified high resistance for non-stomatal deposition which makes this pathway negligible in the model. However, several studies have shown this process to be comparable in magnitude to the stomatal uptake, especially during the night over moist surfaces. Hence, we present here a revised dry deposition in EMAC including meteorological adjustment factors for stomatal closure and an explicit cuticular pathway. These modifications for the three stomatal stress functions have been included in the newly developed MESSy submodel VERTEX, i.e. a process model describing the vertical exchange in the atmospheric boundary layer, which will be evaluated for the first time here. The scheme is limited by a small number of different surface types and generalised parameters. The MESSy submodel describing the dry deposition of trace gases and aerosols (DDEP) has been revised accordingly. The comparison of the simulation results with measurement data at four sites shows that the new scheme enables a more realistic representation of dry deposition. However, the representation is strongly limited by the local meteorology. In total, the changes increase the dry deposition velocity of ozone up to a factor of 2 globally, whereby the highest impact arises from the inclusion of cuticular uptake, especially over moist surfaces. This corresponds to a 6 % increase of global annual dry deposition loss of ozone resulting globally in a slight decrease of ground-level ozone but a regional decrease of up to 25 %. The change of ozone dry deposition is also reasoned by the altered loss of ozone precursors. Thus, the revision of the process parameterisation as documented here has among others the potential to significantly reduce the overestimation of tropospheric ozone in global models.

# 1 Introduction

Ground-level ozone is a secondary air pollutant which is harmful for humans and ecosystems. Besides chemical destruction, a large fraction of it is removed by dry deposition which accounts for about 20 % of the total $O_3$ loss (Young et al., 2018). The process description of dry deposition considers boundary-layer meteorology (e.g. turbulence), chemical properties of the trace gases and surface types. In most global models, dry deposition of trace gases is parameterised using the "resistance in series" analogy by Wesely (1989). The largest deposition rates of ozone occur over dense vegetation (Hardacre et al., 2015) where it mainly follows two pathways, through leaf openings (stomata) and to leaf waxes (cuticle) (Fares et al., 2012). Thereby, stomatal uptake is commonly parameterised following the empirical multiplicative approach by Jarvis (1976) which uses a predefined minimum resistance and multiple environmental response factors like in Zhang et al. (2003); Simpson et al. (2012); Emberson et al. (2000). More advanced formulations often used by land surface models (Ran et al., 2017; Val Martin et al., 2014) are based on the $CO_2$ assimilation by plants during photosynthesis (Ball et al., 1987; Collatz et al., 1992). Both approaches rely on the choice and constraints of ecosystem dependent parameters and have different advantages (Lu, 2018). A further role in coupling stomata to ecosystems play stomatal optimization models whereas optimal stomatal activity with a maximum amount of carbon gain and a minimum loss of water is calculated based on eco-physiological processes (e.g., Cowan and Farquhar, 1977). Of particular interest are stomatal optimization models which, based on eco-physiological processes, maximize carbon gain while minimizing water loss. According to Wang et al. (2020) these models are promising in representing stomatal behaviour and improving carbon cycle modelling. Non-stomatal deposition has been less investigated by now therefore most models use predefined constant resistances or scale it with leaf area index (e.g., Val Martin et al., 2014; Simpson et al., 2012) while some apply an explicit parametrization based on the observational findings of enhance cuticular uptake under leaf surface wetness (Altimir et al., 2006).

The different parametrisations of the (surface) resistances cause main model uncertainties in computing dry deposition fluxes of trace gases, which depend on the response to hydroclimate and land-type specific properties (Hardacre et al., 2015; Wu et al., 2018; Wesely and Hicks, 2000). Thereby, it has been shown that the original Wesely-based parametrisation generally captures well the seasonal and diurnal cycle of dry deposition velocity whereas model-observation discrepancy at seasonal scales arises from biased land type and leaf area index input data (Silva and Heald, 2018). Wong et al. (2019) stated that discrepancies of up to 8 ppb in ground-level ozone arise from different parametrisations.

The current dry deposition scheme of EMAC uses 6 surface types where the parameterised processes represent the forest canopy as a whole (big-leaf approach). Thereby, the uptake over vegetation relies on stomatal deposition as the only pathway determined by the photosynthetically active radiation (Kerkweg et al., 2006). According to Fares et al. (2012) and Rannik et al. (2012) the stomatal uptake in parametrisations often lacks the dependence on meteorological and environmental variables (leaf area index, temperature, vapour pressure deficit). Moreover, several studies (e.g., Hogg et al., 2007; Fares et al., 2012; Clifton et al., 2017) found the contribution of an additional process to dry deposition at the leaf covering of plants. Zhang et al. (2002) firstly derived a parametrisation from field studies which establishes the important link of this process to meteorology. In general, findings by Solberg et al. (2008); Andersson and Engardt (2010); Wong et al. (2019) highlight the importance of

considering the dry deposition-meteorology dependence in global models. Such an extension would realistically enhance the sensitivity of dry deposition to climate variability and would result in a more accurate prediction of ground-level ozone.

Given the importance of ozone as a major tropospheric oxidant, air pollutant and greenhouse gas, an accurate representation of dry deposition is desirable (Jacob and Winner, 2009). Additionally, the significance of a realistic representation of land-atmosphere feedbacks rises in the light of the changing Earth's climate with projected increase of extreme events frequency and intensity (Coumou and Rahmstorf, 2012).

Here, we present a revision of the existing Wesely's based dry deposition scheme in MESSy which has a very simplified representation of vegetation and soil. The modifications are done by well-established findings about the controls of stomatal and cuticular uptake of trace gases. The calculation of stomatal deposition fluxes is extended by including the vegetation density, two meteorological adjustment factors and an improved soil moisture availability function for plant stomata following the multiplicative algorithm by Jarvis (1976). For the first time in MESSy, a parametrisation for cuticular dry deposition dependent on important meteorological and environmental variables is implemented explicitly (Zhang et al., 2003). In Sect. 2, a description of the model set up and the simulations is provided whereas especially the transition to the new vertical exchange scheme is described in detail. Subsequently, the new scheme VERTEX is evaluated. In Sect. 4, the impact of the changes on ozone dry deposition is evaluated on daily and seasonal scales by comparison with measurements at four different sites. Here, advantages, uncertainties and missing processes in the revised scheme are identified. Next, the global impact on ground-level ozone is assessed by separating the effect of the different implemented parametrisations. Then, Sect. 6 provides a description of the uncertainties in modelling stomatal conductance and Sect. 7 comprises an investigation of the sensitivity to model resolution. Sect. 8 summarises the main findings and the remaining process and model uncertainties which form the basis for the provided recommendations. Sect.9 describes future planned developements.

## 2 Model description

This study uses the Atmospheric Chemistry Model ECHAM/MESSy. The Modular Earth Submodel System (MESSy v2.54) (Jöckel et al., 2010) provides a flexible infrastructure for coupling processes to build comprehensive Earth System Models (ESMs) and is utilised here with the fifth generation European Centre Hamburg general circulation model (Roeckner et al. (2003), ECHAM5) as atmospheric general circulation model. The dry deposition process of gases is calculated within the submodel DDEP (Kerkweg et al., 2006). This is described in Section 2.2. It relies on the vertical exchange submodel VERTEX (Sect. 2.1), former E5VDIFF, which contains the calculation of stomatal uptake (Eq. 5) and soil moisture stress (Eq. 12). The stomatal uptake parametrisation is the base for the evapotranspiration scheme in VERTEX (Appendix B) which also incorporates the soil moisture stress.

### 2.1 The new vertical exchange submodel VERTEX

The submodel VERTEX represents land-atmosphere exchange and vertical diffusion as an alternative to the default submodel E5VDIFF in ECHAM5/MESSy. In 2016 Huug Ouwersloot branched VERTEX off from E5VDIFF. He optimised the code

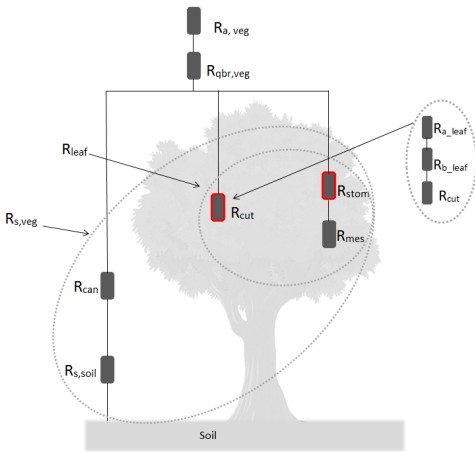

**Figure 1.** Dry deposition resistance analogy, modified resistors are marked with red boxes (adapted from Zhang et al. (2003)).

and applied bug fixes. This includes changes in calculation of the transfer coefficients for vertical diffusion, the latent heat vaporisation, the convective transfer coefficient, the storage of the friction velocity, the roughness length over sea, the kinematic

heat and moisture fluxes and the 2 m and 10 m friction velocity. A detailed description can be found in the Supplement.

## 2.2 Dry deposition over vegetation

Dry deposition of trace gases to vegetation is calculated according to the multiple resistance scheme by Wesely (1989) shown in Figure 1. The scheme, originally designed for a regional model with 11 land types and 5 seasonal categories, is used here with 6 generalized land types (Kerkweg et al., 2006). This was adapted by Ganzeveld and Lelieveld (1995) to the surface scheme

of the ECHAM climate model (Klimarechenzentrum et al., 1992). The vegetation canopy is represented as one system , i.e., the detailed structure and plant characteristics are neglected (one big-leaf approach). Only one assumption about the canopy structure is made: the leaves are horizontally oriented and the leaf density is uniformly vertically distributed (Sellers, 1985). This is required in the formula for the calculation of stomatal resistance (Eq. 5).

The resistances (in $\mathrm{s\,m^{-1}}$) in the big-leaf approach account for mass and energy transfer mainly exerted by the boundary

layer turbulence ($R_a$), molecular diffusion via the quasi-laminar boundary layer ($R_{qbr}$) and heterogeneous losses at the surface ($R_s$) (Kerkweg et al., 2006). With these, the dry deposition velocity $v_d$ of a trace gas $X$ (in $\mathrm{s\,m^{-1}}$) is defined as follows:

$$v_d(X) = \frac{1}{R_a + R_{qbr}(X) + R_s(X)} \tag{1}$$

The dry deposition flux $f_d(X)$ (in $\mathrm{molecules\,m^{-2}\,s^{-1}}$) is determined by multiplying the dry deposition velocity with the trace gas concentration $C(X)$ (in $\mathrm{molecules\,m^{-3}}$):

$$f_d(X) = -v_d(X) \cdot C(X) \tag{2}$$

The total resistance over land combines the resistances over snow, soil, vegetation ($veg$) and wet skin ($ws$) weighted by the respective land covered fraction of a grid box (Kerkweg et al., 2006). In the following, only the latter two are considered. The resistances $R_a$ and $R_{qbr}$ are commonly parameterised with standard formulations from micro-meteorology (Kerkweg et al., 2006; Wesely and Hicks, 1977). For the surface resistance over vegetation ($R_{s,veg}$) the parametrisation according to Zhang et al. (2003) is used:

$$\frac{1}{R_{s,veg}(X)} = \frac{1}{R_{can} + R_{s,soil}(X)} + \underbrace{\frac{1}{R_{cut}(X)} + \frac{1}{R_{stom,corr}(X) + R_{mes}(X)}}_{R_{leaf}(X)} \tag{3}$$

which consists of the soil resistance ($R_{s,soil}(X)$), the in-canopy aerodynamic resistance ($R_{can}$) (as in Kerkweg et al. (2006)) and the leaf resistance ($R_{leaf}(X)$). The gas uptake by leaves ($leaf$) can be separated in two parallel pathways: the cuticular ($cut$) and the stomatal ($stom$) with its associated mesophyllic pathway ($mes$), where the latter has negligible resistance for ozone and highly soluble species (Wesely, 1989). In contrast to the default formulation in MESSy (Eq. A1), the resistances in the updated scheme are provided at canopy scale in order to avoid linear scaling with the Leaf Area Index (LAI, area of leaves [$m^2$]/surface area [$m^2$]). In fact, the linear scaling of resistances with LAI assumes that the leaves act in parallel and overestimates the uptake for high LAI values (>3-4) (Ganzeveld et al., 1998; Baldocchi et al., 1987). Furthermore, the quasi-laminar boundary resistance of individual leaves is included through the cuticular deposition scheme (see Sect. 2.2.2) whereas $R_{qbr,veg}$ is a separate term in the old formulation (Eq. A1).

Due to the importance of stomatal and cuticular uptake for ozone dry deposition their respective parametrisations are modified in this study (see Sec. 2.2.1 and Sec. 2.2.2). Also, ozone deposition to soil might be an important pathway (Schwede et al., 2011; Fares et al., 2012) but process understanding remains limited due to scant observational constraints (Clifton et al., 2020b, a). Stella et al. (2011) showed an exponential increase of soil resistance with surface relative humidity in three agricultural datasets which, however, varies much between different sites (Stella et al., 2019) and contradicts with previous findings (Altimir et al., 2006; Lamaud et al., 2002; Zhang et al., 2002). Models by e.g. Mészáros et al. (2009); Lamaud et al. (2009) apply a linear dependence on soil water content for parameterising soil resistance. These parametrisations rely on input variables like the minimum soil resistance (Stella et al., 2011) which introduce an uncertainty due to measurement constraints. Also, the performance of a mechanistic model as proposed by Clifton et al. (2020b) depends on many input variables and parameters whose estimation is challenging and mostly biome-dependent. Due to these uncertainties and limitations the current parametrization of soil resistance in MESSy (see Kerkweg et al. (2006) for details) was not modified in this study.

### 2.2.1 Uptake through plant stomata

The stomata are actively regulated openings between the plant cells. They are scattered mostly over the lower (hypostomatous) epidermis of leaves. They control the $H_2O$ and $CO_2$ exchange by plants which is the essential coupling of vegetation to the atmosphere and therefore to weather and climate. Here, the default parametrisation of stomatal resistance (Eq. A2) is extended by adding dependencies on meteorological variables according to the Simple Biosphere Model (SiB) by Sellers et al. (1986)

based on previous work by Jarvis (1976) for temperature (T) and vapour pressure deficit (VPD):

$$R_{stom,corr}(X) = \frac{R_{stom}(PAR, LAI)}{f(W_s) \cdot f(T) \cdot f(VPD)} \cdot \frac{D_{H_2O}}{D(X)} \tag{4}$$

The optimal stomatal resistance for water ($R_{stom}(PAR, LAI)$) is corrected with the ratio of the molecular diffusivity of the species ($D(X)$) and water ($D_{H_2 0}$). The optimal stomatal resistance depends on the photosynthetically active radiation (PAR) and Leaf Area Index (LAI) (Ganzeveld and Lelieveld, 1995; Sellers, 1985):

$$R_{stom}(PAR, LAI) = \frac{kc}{\left[ \frac{b}{dPAR} ln \left( \frac{d \exp(kLAI)+1}{d+1} \right) - ln \left( \frac{d+\exp(-kLAI)}{d+1} \right) \right]} \tag{5}$$

where k $= 0.9$ is the extinction coefficient, c $= 100 \, \mathrm{s\,m^{-1}}$ is the minimum stomatal resistance and $a = 5000 \, \mathrm{J\,m^{-3}}$, $b = 10$ $\mathrm{W\,m^{-2}}$ and $d = \frac{a+b \cdot c}{c \cdot PAR}$ are fitting parameters (Sellers, 1985). For historical reasons, LAI was set to 1 in order to obtain the stomatal resistance at leaf level (Ganzeveld and Lelieveld, 1995). This has been changed and the seasonal evolution of stomatal resistance now follows the LAI which, in our study, is based on a 5-year climatology of monthly Normalised Differential Vegetation Index (NDVI) satellite data (Ganzeveld et al., 2002).

First, the stomatal resistance is corrected by the inverse of the temperature stress factor ($1/f(T)$) derived by Jarvis (1976):

$$f(T) = b_3(T - T_l)(T_h - T)^{b_4} \tag{6}$$

$$b_3 = (T_0 - T_l)(T_h - T_0)^{-b_4} \tag{7}$$

$$b_4 = (T_h - T_0)/(T_h - T_l) \tag{8}$$

where the empirical parameters are $T_h = 318.15$ K, $T_l = 268.15$ K and $T_0 = 298.15$ K.

Secondly, following the analysis by Katul et al. (2009), a stress factor dependent on vapour pressure deficit ($1/f(VPD)$) was added to the calculation of stomatal resistance in VERTEX:

$$p_{H_2O,sat}(T) = 0.61078 \exp \left( \frac{17.1 \cdot T(p_{H_2O})}{235 + T(p_{H_2O})} \right) \tag{9}$$

$$VPD = p_{H_2O,sat}(T) - p_{H_2O} = \left( 1 - \frac{RH}{100} \right) p_{H_2O,sat}(T) \tag{10}$$

$$f(VPD) = VPD^{-\frac{1}{2}} \tag{11}$$

with $T(p_{H_2O})$ (in K) as the surface temperature, $p_{H_2O}$ (in kPa) as the pressure of water vapour and $p_{H_2O}(T)$ [kPa] the pressure of saturated air. The vapour pressure deficit is calculated according to Kraus (2007).

While the stomatal resistance at canopy scale is actually calculated within the MESSy submodel VERTEX, the submodel DDEP uses it for the calculation of dry deposition fluxes. Thus, in DDEP the user can choose between the old scheme based on Ganzeveld and Lelieveld (1995) and the new scheme actually using the stomatal resistance at canopy scale. The latter is activated by setting the DDEP $\&CTRL$ namelist parameter $l\_ganzeori$ to $.FALSE.$. How the stomatal resistance is calculated is chosen in VERTEX by the $\&CTRL$ namelist parameter $irstom$.

  – $irstom = 0$ activates the original parametrisation.

- Separate modifications:

    - $irstom = 2$: variable $LAI$,

    - $irstom = 3$: T dependency,

    - $irstom = 4$: VPD dependency, respectively.

- $irstom = 5$: all modifications.

- $irstom = 1$: stomatal resistance with variable $LAI$ at leaf scale. Instead of choosing LAI=1 in Eq. 5 to represent the stomatal resistance at leaf level, as is done by the original code, Eq. (5) is calculated at canopy level using the actual LAI and then multiplied by LAI to obtain the average stomatal resistance at leaf level. For this case, the DDEP namelist parameter $l\_ganzeori$ have to be set to .TRUE.

The stomatal activity of plants and the strength of surface-atmosphere coupling strongly depend on the parameterised plant-water stress (Combe et al., 2016). The soil water budget is represented by a "bucket scheme" where the soil water in a single layer is prescribed by a geographically varying predefined field capacity and soil wetness governed by transpiration, precipitation, runoff, snow melt and drainage (Roeckner et al., 2003). This scheme is used by so called "first-generation" models. However, EMAC controls evapotranspiration through the stomatal resistance (Appendix B) which is the most important fea-

ture of biophysical ("second-generation") land-surface models. Thereby, the stomatal resistance is calculated often like the one described here (Eq. 4) including temperature, VPD and soil moisture stress (Seneviratne et al., 2010; Sellers et al., 1997). The originally used plant-water stress function of Jarvis (1976) and Sellers et al. (1986), however, relies on leaf water potential ($f(\psi)$) for different plant types, which is difficult to estimate. Hence, EMAC uses a plant-water stress function dependent on soil moisture ($f(W_s)$). The default parametrisation (Eq. A, $ifws = 0$ in VERTEX $\&CTRL$) applies as lower threshold the

permanent wilting point of plants ($W_{pwp}$, 35% of field capacity[1]) in the calculation of the soil moisture stress factor ($f(W_s)$). However, soil moisture is significantly underpredicted by the model in some regions and the calculated $f(W_s)$ can be 0 for long periods. This is unrealistic and effectively shuts down dry deposition, e.g. during the dry season in the Amazon region. For this reason $f(W_s)$ is parameterised here according to the original formulation by Delworth and Manabe (1988) removing the lower limit:

$$f(W_s) = \begin{cases} 1 & W_s(t) > W_{cr} \\ \frac{W_s(t)}{W_{cr}} & W_s(t) \leq W_{cr} \end{cases} \tag{12}$$

where $W_s(t)$ is the surface soil wetness (in m). $W_{cr}$ (in m) is defined as the critical soil moisture level (75 % of the field capacity) at which the transpiration of plants is reduced. The modified parametrisation in Eq. 12 can be applied by setting the $\&CTRL$ parameter $ifws = 1$ in the VERTEX namelist.

---

[1]maximum amount of water the soil can hold against gravity over periods of several days

### 2.2.2 Cuticular deposition

According to several field studies (e.g., Van Pul and Jacobs, 1994; Hogg et al., 2007; Fares et al., 2012) cuticular deposition is an important contributor to ozone uptake and should not be neglected in models. Therefore, an explicit parametrisation of cuticular deposition as used in many North American air quality modelling studies (Huang et al., 2016; Kharol et al., 2018) has been implemented. The gas uptake by leaf surfaces is based on two parallel routes, for which an analogy to ozone (highly reactive) and sulphur dioxide (very soluble) is used. The cuticular resistance is calculated as:

$$R_{cut}(X) = \frac{R_{cut,d}(O_3)}{10^{-5} \cdot H(X) + s_{reac}(X)} \tag{13}$$

where $H(X)$ is the effective Henry's law coefficient as measure for the solubility. The reactivity of a species is rated by the parameter $s_{reac}$. For highly reactive species ($s_{reac} = 1$) the same property as for ozone is assumed (second term in Eq. 13), while for less reactive species ($s_{reac} = 0.1, 0$) the uptake is effectively reduced (Wesely, 1989). For soluble species, the uptake at wet skin is assumed to be similar to the one of sulphur dioxide and is calculated as:

$$R_{ws}(X) = \left[ \frac{1/3}{R_{cut,w}(SO_2)} + 10^{-7} \cdot H(X) + \frac{s_{reac}(X)}{R_{cut,w}(O_3)} \right]^{-1} \tag{14}$$

where $R_{cut,w}(SO_2)$ and $R_{cut,w}(O_3)$ are the resistances of sulphur dioxide and ozone at wet surfaces, respectively. The constant values of the default formulae (Eq. A4, A5) are replaced by parametrisations which account for the meteorological dependence of cuticular uptake according to Zhang et al. (2002):

$$R_{cut,d}(O_3/SO_2) = \frac{R_{cut,d0}(O_3/SO_2)}{\exp(0.03 \cdot RH) \cdot LAI^{0.25} \cdot u_*} \tag{15}$$

$$R_{cut,w}(O_3/SO_2) = \frac{R_{cut,w0}(O_3/SO_2)}{LAI^{0.5} \cdot u_*} \tag{16}$$

where the cuticular resistance of $O_3$ and $SO_2$, respectively, is distinguished for dry canopies ($R_{cut,d}$) and wet canopies ($R_{cut,w}$) depending on relative humidity ($RH$ in %), Leaf Area Index ($LAI$ in $m^2\,m^{-2}$) and friction velocity ($u_*$ in $m\,s^{-1}$). The input parameters are $R_{cut,d0}(O_3)$=5000 s m$^{-1}$, $R_{cut,w0}(O_3)$=300 s m$^{-1}$ and $R_{cut,d0}(SO_2)$=2000 s m$^{-1}$ (Zhang et al., 2002). For rain and dew conditions, values of 50 s m$^{-1}$ and 100 s m$^{-1}$ are prescribed for $R_{cut,w0}(SO_2)$. In contrast to traditional approaches,
these parametrisations also consider the aerodynamic and the quasi-laminar boundary resistances of individual leaves. For the usage in MESSy this can be switched on via $l\_ganzeori = .FALSE.$ in the $\&CTRL$ namelist of DDEP.

### 2.3 Simulations

In order to answer the different research questions of this study, two different types of simulations have been performed (Tab. 1):

(1) Simulations to investigate dry deposition and the effect of the modifications in VERTEX:
These simulations are based on the Chemistry-Climate Model Initiative (CCMI) setup (Jöckel et al., 2016). To allow for comparison with measurements, the model dynamics have been nudged towards realistic meteorology by the assimilation of data

**Table 1.** List of EMAC simulations

| Simulation | Spatial resolution | Time period | Remarks |
|---|---|---|---|
| **(1) Dry deposition mechanism: CCMI chemistry, nudged, no feedbacks (QCTM)** | | | |
| REST42 | T42L31 (2.8° x2.8°) | 2009/2010 | irstom=5, ifws=1, l_ganzeori=F |
| REST63 | T63L31 (1.9° x1.9°) | 2009/2010 | irstom=5, ifws=1, l_ganzeori=F |
| REV (revised) | T106L31 (1.1° x1.1°) | 2009-2015, 2017-June 2018 | irstom=5, ifws=1, l_ganzeori=F |
| DEF (default) | T106L31 (1.1° x1.1°) | 2009-2015, 2017-June 2018 | default ddep scheme |
| REV-fws | T106L31 (1.1° x1.1°) | 2009/2010 | irstom=5, ifws=0, l_ganzeori=F |
| REV-fTfD | T106L31 (1.1° x1.1°) | 2009/2010 | irstom=2, ifws=1, l_ganzeori=F |
| REV-NNTR | T106L31 (1.1° x1.1°) | 2014/2015 | free-running, all ddep modifications (as REV), all stress factors applied to evapotranspiration (izwet=1). |
| **(2) Climatology comparison: no chemistry, free-running** | | | |
| clim-E5 | T42L90 (2.8° x2.8°, up to 0.01 hPa) | 1979-2008 | E5VDIFF for vertical exchange |
| clim-VER | T42L90 (2.8° x2.8°, up to 0.01 hPa) | 1979-2008 | VERTEX for vertical exchange |

from the European Centre for Medium-range Weather Forecasting (ECMWF) (Jöckel et al., 2010). Additionally, the QCTM mode is used, i.e., the chemistry does not feed back to the dynamics, resulting in the same meteorology for all simulations (Deckert et al., 2011). All modifications for the dry deposition scheme are employed in a 7-year simulation (REV, 2009-2015). Additionally, a 1.5-year simulation covering the period 2017 to July 2018 (2017 as spin-up) has been performed to cover the measurement periods (Sect. 4). For the same periods simulations with the same configuration except applying the default dry deposition scheme (DEF) have been conducted. The individual effects of the different modifications are investigated by two 2-year simulations employing the different namelist switches (Sect. 2.2). Moreover, a free-running sensitivity simulation with an additional temperature and drought stress factor for evapotranspiration (Appendix B) has been performed aiming at an improved representation of local meteorology especially in the Amazon. The station simulation output and the global output are analysed in Sections 4 and 5, respectively. In addition, two 2-year simulations are realised for different horizontal resolutions (REST42, REST63) to investigate the resolution dependency of dry deposition (Sect. 7). All these simulations use 31 model layers with the top at 10 hPa and take the first year of simulation as spin-off.

(2) Simulations for the evaluation of VERTEX as boundary layer scheme:

Two pure dynamical (i.e., without chemistry) 30-year simulations with the old (clim-E5) and the new boundary layer description (clim-VER), respectively, have been performed.

All simulations were performed at the Jülich Supercomputing Center with the JURECA Cluster (Jülich Supercomputing Centre, 2018).

**Table 2.** Overview of tuning parameter settings and global mean properties

| Parameters | | EMAC(E5VDIFF) | EMAC(VERTEX) |
|---|---|---|---|
| Cloud mass-flux above level of non-buoyancy | | 0.3 | 0.3 |
| Entrainment rate for shallow convection | | $1e-3$ | $1e-3$ |
| Entrainment rate for deep convection | | $1e-4$ | $1e-4$ |
| Conversion rate to rain in convective clouds | | $1.5e-4$ | $1.6e-4$ |
| **Properties** | **Observed** [a] | **EMAC(E5VDIFF)** | **EMAC(VERTEX)** |
| Total cloud cover [%] | | 67.12 | 67.27 |
| Water vapour path [$\mathrm{kg\,m^{-2}}$] | | 25.03 | 24.83 |
| Liquid water path [$\mathrm{kg\,m^{-2}}$] | | 0.077 | 0.077 |
| Total precipitation [mm/d] | | 1.28 | 1.31 |
| Surface net shortwave [$\mathrm{W\,m^{-2}}$] | 152-167 | 158.27 | 158.32 |
| Surface net longwave [$\mathrm{W\,m^{-2}}$] | -(40-57) | -54.82 | -54.93 |
| Surface sensible heat flux [$\mathrm{W\,m^{-2}}$] | -(16-19) | -18.75 | -19.65 |
| Surface latent heat flux [$\mathrm{W\,m^{-2}}$] | -(75-87) | -87.45 | -88.73 |
| Planetary albedo [%] | | 32.38 | 32.37 |
| Shortwave net at TOA [$\mathrm{W\,m^{-2}}$] | 238-244 | 230.99 | 231.00 |
| Longwave net at TOA [$\mathrm{W\,m^{-2}}$] | -(237-241) | -232.46 | -232.55 |
| Radiation imbalance at TOA [$\mathrm{W\,m^{-2}}$] | | -1.47 | -1.55 |

[a]Stevens and Schwartz (2012)

## 3 VERTEX evaluation

In order to advise the usage of VERTEX (with the default settings) as the default vertical exchange submodel in MESSy the dynamics produced by both submodels are compared. Therefore, two dynamical, free running, 30-year simulations have been performed using the E5VDIFF or the VERTEX submodels, respectively. To obtain a comparable radiative imbalance at TOA (top of the atmosphere) with VERTEX the four cloud parameters have been tuned in advance according to Mauritsen et al. (2012). The tuning factors can be found in Table 2. The radiative imbalance at TOA is slightly positive at present-day conditions (Mauritsen et al., 2012; Stephens et al., 2012), here E5VDIFF gives a negative value. The difference between the tuned VERTEX and E5VDIFF is small and within the uncertainty range of $\pm 0.4\ \mathrm{W\,m^{-2}}$.

Additionally, global mean values of surface temperature, cloud liquid water, relative humidity and planetary boundary layer height of EMAC using E5VDIFF and EMAC using VERTEX with the respective uncertainty range for the period 1979-2008 are represented in Figure 2. The results for cloud liquid water and planetary boundary height show no significant differences between the VERTEX and E5VDIFF simulation since each annual means falls in the confidence interval of the other. This

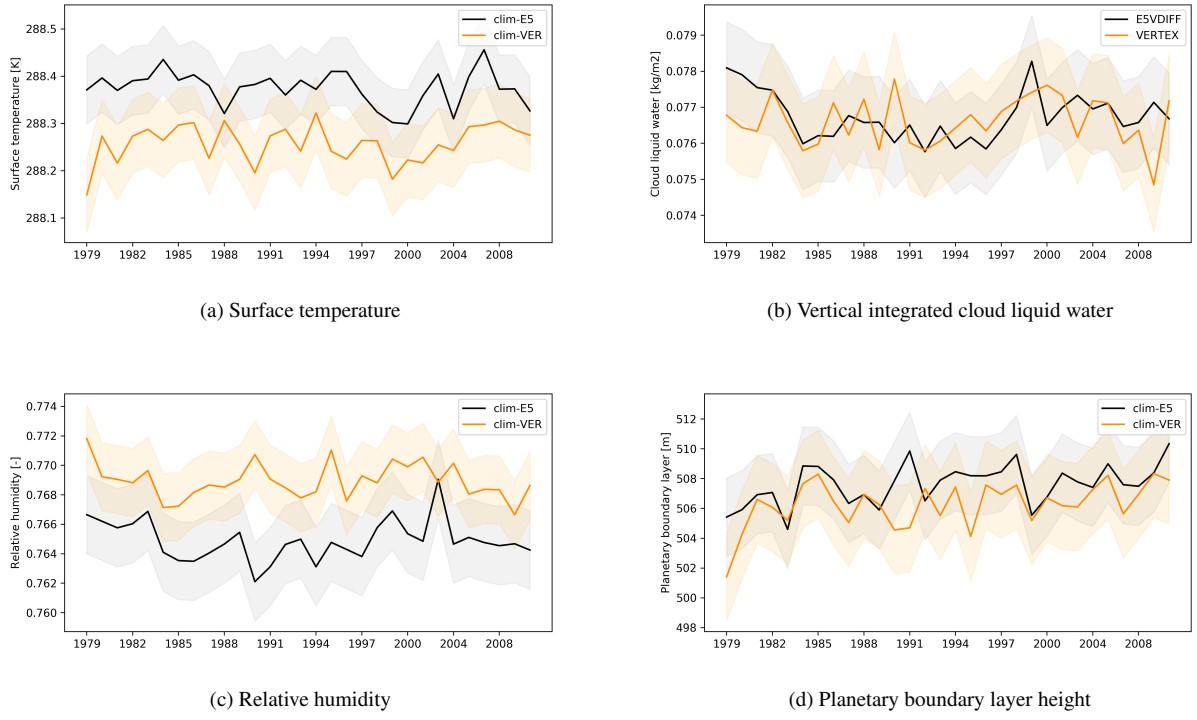

(a) Surface temperature

(b) Vertical integrated cloud liquid water

(c) Relative humidity

(d) Planetary boundary layer height

**Figure 2.** Global mean properties and the uncertainty range (95. confidence interval in shaded) of the climatology simulations with E5VDIFF (clim-E5) and with VERTEX (clim-VER) for the period 1979-2008.

is not always the case for surface temperature and relative humidity. However, the 30-year means of surface temperature and relative humidity simulated by E5VDIFF and VERTEX are not significantly different.

## 4 Evaluation with deposition measurements

To assess the impact of the code revision/modifications on the variability of dry deposition we compare the sensitivity simulations DEF, REV, REV-fTfVPD, REV-fws and REV-NNTR (see Tab. 1, all at T106L31 resolution) with dry deposition measurements at four field sites (listed in Table 3). The chosen data sets are the best available of ozone dry deposition (flux data and ozone mixing ratio or velocity data) with the required temporal resolution and coverage of diverse biomes of the world. The analysis is aimed at covering the recent decade which includes the most extreme drought and heat events (where

the stomatal stress factors are aimed for). For the reason of uniqueness and importance of atmospheric processes in a remote and pristine forest like the Amazon Basin we included measurements from there among others the Amazonian Tall Tower Observatory (ATTO). Ozone dry deposition fluxes were measured with the eddy covariance and gradient method (Ontario). From this data, deposition velocities were calculated by the means of ozone concentration data. The eddy covariance technique

**Table 3.** Dry deposition measurements. In the description of vegetation/climate the reported Leaf Area Index (LAI, in $m^2\,m^{-2}$) is given in brackets, $v_d^{mod}$ and $v_d^{obs}$ are the average measured and modelled dry deposition velocity.

| Site | Vegetation/climate | Location (height) | Time period | $v_d^{mod}(v_d^{obs})$ cm s$^{-1}$ | Reference |
|---|---|---|---|---|---|
| Hyytiälä, South-ern Finland (SMEARII) | boreal forest, Scots Pine, (LAI=3-4)/ cold temperate | 61.85N 24.28E (22 m/16 m[a]) | 2010-2012 | 0.29 (0.28) | Keronen et al. (2003) |
| Lindcove research station, California (US) | Citrus Orchard (LAI=3)/ Mediterranean | 36.35N 119.09W (131 m) | Oct.2009-Nov.2010 | 0.22 (0.49) | Fares et al. (2012) and Fares[b] |
| Borden research station, Ontario, Canada | mixed forest (LAI=4.6)/ temperate | 44.19N 79.56W (33 m) | 2010-2012 | 0.34 (0.47) | Wu et al. (2018) |
| Amazonian Tall Tower (ATTO), Manaus, Brazil | rainforest (LAI=6)/ tropical humid | 2.15S -59.01W (41 m) | November 2015, April/May 2018 | 0.18 (0.67), 0.33 (1,0) | available on request: Matthias Sörgel (m.soergel@mpic.de) |

[a]Meteorological measurement height

[b]Ozone data is not available here

determines a turbulent flux by the covariance of the measured vertical velocity and the gas concentration. Due to the stochastic nature of turbulence, these measurements have an uncertainty of 10 to 20 % under typical observation conditions (Rannik et al., 2016). For the gradient method used at Borden forest research station the dry deposition flux was estimated from concentration gradients below and above the canopy and the eddy diffusivity according to the Monin-Obukhov similarity theory. The estimated dry deposition velocities ($V_d$) show an uncertainty of $\approx 20$ % which is due to the assigned canopy, the inherent limitations of the algorithm and the measurement uncertainties in concentrations. However, results are in good agreement with other eddy covariance measurements (Wu et al., 2016).

## 4.1 Annual cycle of dry deposition

The annual cycle of dry deposition is mainly driven by the evolution of vegetation and is generally represented well in models (Silva and Heald, 2018). We use here the long time series measured at Borden and Hyytiälä to identify the impact of the code modifications on the annual cycle of dry deposition velocity. The available micro-meteorological data help to distinguish the different effects. From the hourly data, we calculated multiyear (2010-2012) monthly means. To explore the contribution of stomatal and cuticular uptake, the individual velocities are calculated for O$_3$ according to the model calculations (Kerkweg

et al., 2006):

$$G_{cut,d} = \frac{(1-ws) \cdot (1-cvs) \cdot veg}{R_{cut,d}(\text{O}_3)} \qquad G_{cut,w} = \frac{ws \cdot (1-cvs)}{R_{cut,w}(\text{O}_3)} \tag{17}$$

$$G_{ns} = G_{cut,d} + G_{cut,w} \tag{18}$$

$$G_{stom} = \frac{(1-ws) \cdot (1-cvs) \cdot veg}{R_{stom,corr}(\text{O}_3)} \tag{19}$$

$$v_p = \frac{G_p}{G_{stom} + G_{ns}} \cdot v_d \tag{20}$$

where $G$ names the individual conductances (inverse of resistance) of stomata ($stom$), dry cuticle ($cut,d$), wet cuticle ($cut,w$) and non-stomata ($ns$). $veg$, $ws$ and $cvs$ give the vegetation fraction, the wet skin fraction and the snow covered fraction, respectively. $G_p$ and $v_p$ are the individual conductance and the velocity of one pathway. Further terms are described in Sect. 2.2.

The multiyear (2010-2012) annual cycle of the simulated dry deposition velocity at Borden forest (Fig. 3a) captures the observed cycle well until June. The new scheme reproduces the observations better than the old scheme. This is a consequence of the increase in nighttime mean velocities due to the much larger cuticular contribution (Fig. A1a, A1b). However, due to the overestimated stomatal uptake in the default scheme (see Sect. 2.2.1) only slight deviations from the new dry deposition scheme are visible in the daily mean shown in Figure 3a. The mismatch of the simulated and measured $V_d$ from August to

October is a consequence of the underestimation of relative humidity leading to too low simulated cuticular deposition (Fig. 3c, 3e). This effect exceeds the impact of the overestimation of relative humidity (only) in summer, because the LAI is higher in summer. In general, the cuticular uptake parametrisation accounting for LAI, friction velocity, RH and surface wetness conditions performs, in our simulations, better than parametrisations without these dependencies as expected from the study of Wu et al. (2018). Unfortunately, the cuticular uptake parametrisation also introduces uncertainties to the modelled non-stomatal

uptake. Moreover, accounting for biogenic volatile organic compounds (BVOCs) like in Makar et al. (2017) would enhance in-canopy loss of ozone, significantly increase non-stomatal dry deposition and lead to improved simulation results (Wu et al., 2018). The representation of in-canopy air chemistry is outside the scope of the present study but planned within a subsequent study.

In contrast, the amplitude of the annual cycle and the mean of dry deposition fluxes in Hyytiälä are overestimated by both

schemes during spring and summer (Fig. 3b). For the default scheme, this is due to the oversimplification of the stomatal uptake that only accounts for a constant LAI of 1 $\text{m}^2\,\text{m}^{-2}$ (see Sect. 2.2.1) which is far from the measured LAI of 3-4 $\text{m}^2\,\text{m}^{-2}$ during this period (Keronen et al., 2003). Enabling the new scheme (REV), increases the dry deposition velocity which reproduces the measured values in autumn better. The contribution of non-stomatal dry deposition of 25-45 % during day reported by Rannik et al. (2012) is represented partly by that. However, the new scheme leads to an even higher overestimation by the

model from April to July. The sensitivity simulation REV-fws (default $f(W_s)$) points to the increase of soil moisture stress function (see Sect. 2.2.1, Eq. 12) as one reason for the overestimation of $V_d$ in summer (Fig. 3b, A2b). Moreover, the overestimation in June/July is partly ($\sim 10$ %) due to the too high model LAI compared to the measured values of 3-4 (Fig. A2a). The remaining gap (Fig. 3f) can be explained by restricting the analysis to wet conditions (RH >70 %) only, and the analysis of the sensitivity simulation REF-fTfD (no $f(T)$ and $f(VPD)$). This suggests that the overestimated $V_d$ (Fig. A2c) in summer is

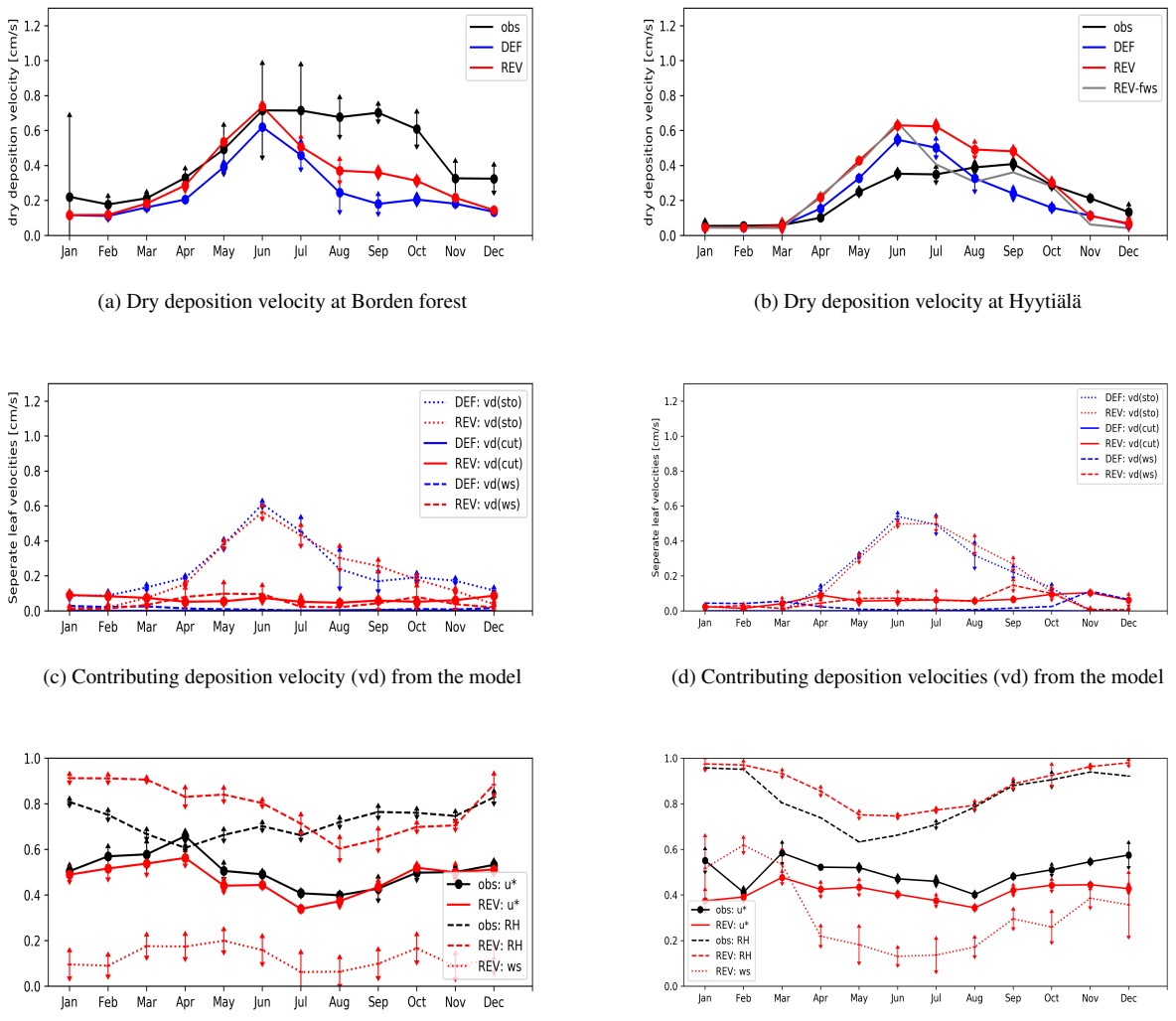

(a) Dry deposition velocity at Borden forest  (b) Dry deposition velocity at Hyytiälä

(c) Contributing deposition velocity (vd) from the model  (d) Contributing deposition velocities (vd) from the model

(e) Friction velocity (u*), Relative humidity (RH), Wet skin fraction (ws)  (f) Friction velocity (u*), Relative humidity (RH), Wet skin fraction (ws)

**Figure 3.** Measured (obs) and modelled (DEF, REV) multiyear mean (2010-2012) and REV-fws (2010) annual cycle. Left: Borden forest, right: Hyytiälä, for (a) and (b) arrows give $1\sigma$

due to the stress factors for stomatal uptake since the modelled and measured temperature mismatch. VPD has been identified by Rannik et al. (2012) as a strong driver of day-time total deposition velocity which confirms the importance of inclusion of VPD dependence for stomatal uptake.

## 4.2 Importance of stress factors for the diurnal variation of deposition

The short-term measurements at Lindcove research station and at Amazonian Tall Tower Observatory (ATTO) are used to assess the impact of the stress factors on the diurnal cycle of dry deposition velocity in spring and summer. Additionally, micro-meteorological and additional flux data make possible to consider the stomatal resistance ($\sim$inverse of the velocity, calculations according to Fares et al. (2012)) and the underlying meteorological conditions. Since the respective micro-meteorological measurements are not available at ATTO, data extracted from the ERA5 global climate reanalysis at the 1000 hPa pressure level (Copernicus, 2017) is used here.

The diurnal cycle of dry deposition velocity at the Lindcove research station follows the solar variation (Fares et al., 2012) and is generally well reproduced by the model with the best match in spring (Fig. 4a). The revised dry deposition scheme reduces the underestimation of measured night-time $V_d$ due to the inclusion of cuticular uptake, which Fares et al. (2012) identified as an important ozone sink for exactly this measurement site. The measured dry deposition velocity increases at sunrise (around 15 UTC) and remains almost constant during the day. This is only reproduced by the revised dry deposition scheme. The comparison of the dry deposition velocity from the revised scheme (red line) and the velocity without stomatal T and VPD stress (gray line) in Figure 4a illustrates the necessity of accounting for the stress factors. This is consistent with Fares et al. (2012) who report a high negative correlation of $V_d(sto)$ with VPD and temperature and relates it to stomatal stress. The direct comparison of the stomatal resistances calculated from measured and modelled variables (Fig. 4c) shows an improvement of the modelled resistances (comparing DEF an REV). However, the modelled daytime stomatal resistance is still too high compared to the measurements. This points to an underestimation of stomatal uptake by the model during day. A small fraction can be explained by the direct effect of the stomatal soil moisture stress in the model which does not occur in reality since the Citrus Orchard was watered during the measurement campaign. Contrastingly in summer, the model underestimation of $V_d$ is higher than in spring (Fig. 4b). As seen from the comparison of stomatal resistance values (Fig. 4d) the model underestimates the stomatal uptake. This is because the irrigation of the Orchard leads to cooling sustained evapotranspiration and keeps $f(T)$ low. Thus in the model, a too high temperature stress act on the stomata. Moreover, neglecting the soil moisture stress on stomata would bring the stomatal resistance values closer since the irrigation at the site ensures a constant and high soil moisture. The irrigation of the Citrus Orchard during day also enhances surface wetness and favours deposition at cuticles (Fares et al., 2012; Altimir et al., 2006) which cannot be captured by the model. Fares et al. (2012) estimate the stomatal contribution to only account for 20-45 % of the total daytime dry deposition flux during both seasons and point to soil deposition and reactions of ozone with NO and VOCs as major sinks at Citrus Orchard, especially during flowering season. The contribution of these pathways is expected to be enhanced by the inclusion of further biogenic VOCs within the chemical mechanism and the explicit parametrisation of in-canopy residence and transport. Tropical forests are known to be effective $O_3$ sinks with observed mean midday maximum dry deposition velocity of 2.3 $\mathrm{cm\,s^{-1}}$ (Rummel et al., 2007) due to much higher LAI compared to other sites (e.g. Lindcove). The measured dry deposition velocity at ATTO shown in Fig. 5a and Fig. 5b is no exception but shows a high variability (standard deviation). The diurnal cycle follows the solar radiation with maximum $V_d$ at 15 UTC and highest amplitude during the wet season (April/May 2018). The amplitude of the diurnal cycle is highly

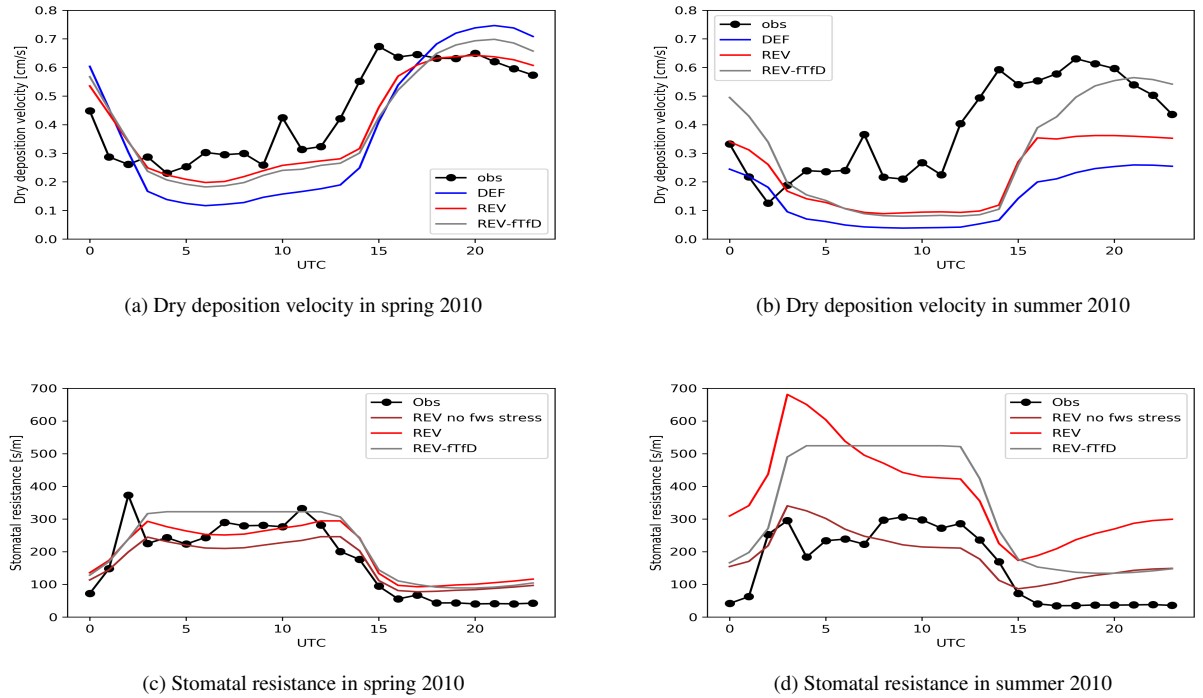

**Figure 4.** Diurnal cycles of measured (obs) and modelled (DEF,REV,REV-fTfD) ozone dry deposition velocity and stomatal resistance in spring and summer 2010 at Lindcove research station.

underestimated in both EMAC simulations with the highest mismatch during daytime. This is similar for other models. In fact, Hardacre et al. (2015) report a general and large underestimation of dry deposition velocities by models over tropical forests with highest predicted values of 0.25 $\mathrm{cm\,s^{-1}}$. Here, the simulation with the revised dry deposition scheme (REV) shows only

a minor increase of $V_d$ during the wet season. Since stomatal uptake is known to be an important daytime sink (Freire et al., 2017), the underestimation of the total dry deposition flux is partly attributed to a too low simulated stomatal uptake caused by the overestimation of temperature and the underestimation of relative humidity (Fig. A3). The increase of dry deposition velocity by the new scheme is mainly due to the lowered soil moisture stress on stomata ($f(W_s)$) shown in Fig. 5e. Freire et al. (2017) also links stomatal uptake to the efficiency of turbulent mixing in transporting ozone down to the canopy. In general,

10 % of the total ozone sink during daytime and 39 % during night is associated with in-canopy processes (Freire et al., 2017). Freire et al. (2017) and Bourtsoukidis et al. (2018) identified the oxidation of sesquiterpenes as an important contributor to the chemical nighttime sink. Cuticular deposition might also play a role in humid conditions during night (Rummel et al., 2007) which is underestimated by the model due to the biased relative humidity (Fig. 5c).

The uncertainty introduced by the mismatching meteorology becomes even more obvious when comparing measurements and

simulations for November 2015. This month was characterised by temperatures of 2 to 3 °C above average and unusual little rainfall (compared to usual conditions in this season) due to a strong El Nino event (National Centers for Environmental Infor-

mation). The dryness is overestimated by the model with a too high temperature ($\Delta$=+5 to +8 K), too low relative humidity ($\Delta$=-30 to -40 %)) and too dry soil. The lack of available soil moisture ($f(W_s)$=0) effectively shuts down stomatal deposition in the default simulation (DEF), whereas the modification of the soil moisture stress function (neglecting the artificial lower limit, see Eq. 12) in the revised model (REV) allows for an increased deposition (Fig. 5b). The temperature and relative humidity biases result in corresponding mismatching stress factors for the stomata that are double the ones derived from reanalysis data (Fig. 5f). This mismatch leads to an underestimation of stomatal uptake. This result is confirmed by the sensitivity simulation REV-NNTR for which no meteorological nudging has been applied and the stress factors $f(T)$ and $f(VPD)$ are also used for the calculation of evapotranspiration. The REV-NNTR simulation yields much more realistic results compared to the measurements capturing at least 50 % of the measured $V_d$ during day (Fig. 5b). This improvement is partly due to the omission of nudging. As the latter can have a detrimental effect on precipitation and evaporation (Jeuken et al., 1996). The temperature bias of the model is associated with the missing soil moisture buffer simulated by the bucket scheme. Incorporating a 5-layer scheme has been shown to lead to a more realistic soil water storage capacity especially in the Amazon and to a removal of this bias (Hagemann and Stacke, 2013). Nevertheless, the REV-NNTR simulation suggests that the stress factors $f(T)$ and $f(VPD)$ significantly contribute to buffer soil moisture and ameliorate the dryness bias.

## 5 Global impact on ground-level ozone

Given the importance of dry deposition for ground-level ozone and the uncertainty of dry deposition parametrisations in models (Young et al., 2018; Hardacre et al., 2015) the global impact of the implemented code changes is assessed in this section.

The global (boreal) summer mean distributions of deposition velocity and ground-level mixing ratio for $O_3$ shown in Figs. 6a/6b are generally in the same range as reported for global models (e.g. Val Martin et al. (2014); Hardacre et al. (2015)). However, like most global models, EMAC overestimates tropospheric ozone in comparison to satellite observations (Righi et al., 2015). Applying the revised dry deposition scheme increases the mean summer $V_d$ by up to 0.5 $\mathrm{cm\,s^{-1}}$ (Fig. 6c). The highest fraction of this increase arises from the inclusion of cuticular uptake at wet surfaces ($V_{cut,w}$) (Fig. A4b). The effect is large over the most northern continental regions (Fig. 6f) and even more pronounced where LAI is high like in Scandinavia and East Canada (for LAI distribution see Fig. A4a). Additionally, the uptake at dry surfaces ($V_{cut,d}$) is enhanced with up to 0.3 $\mathrm{cm\,s^{-1}}$ higher dry deposition velocity (Fig. 6e). This is because the default scheme applies a very high constant resistance for this process.

Concerning the stomatal deposition, the impacts of three different stress factors are considered. First, over relatively dry soil, i.e., where soil moisture exceeds 35 % of field capacity (wilting point of plants), the soil moisture stress is reduced by the modified parametrisation. Neglecting the plants' wilting point as the lower limit for soil moisture stress on stomata weakens the dependency on field capacity. Thus, dry deposition is enhanced by up to 0.32 $\mathrm{cm\,s^{-1}}$ as illustrated in Figure 7a. Second, the inclusion of temperature and (third) VPD adjustment factors, indeed, leads to a spatially varying impact of $\pm$ 0.27 $\mathrm{cm\,s^{-1}}$ change in $V_d$ (Fig. 7b). In humid and cold temperate regions, like Siberia and Canada, no temperature stress appears and the VPD adjustment factor increases the stomatal uptake. In East U.S., Kazakhstan and Central Amazon during boreal summer

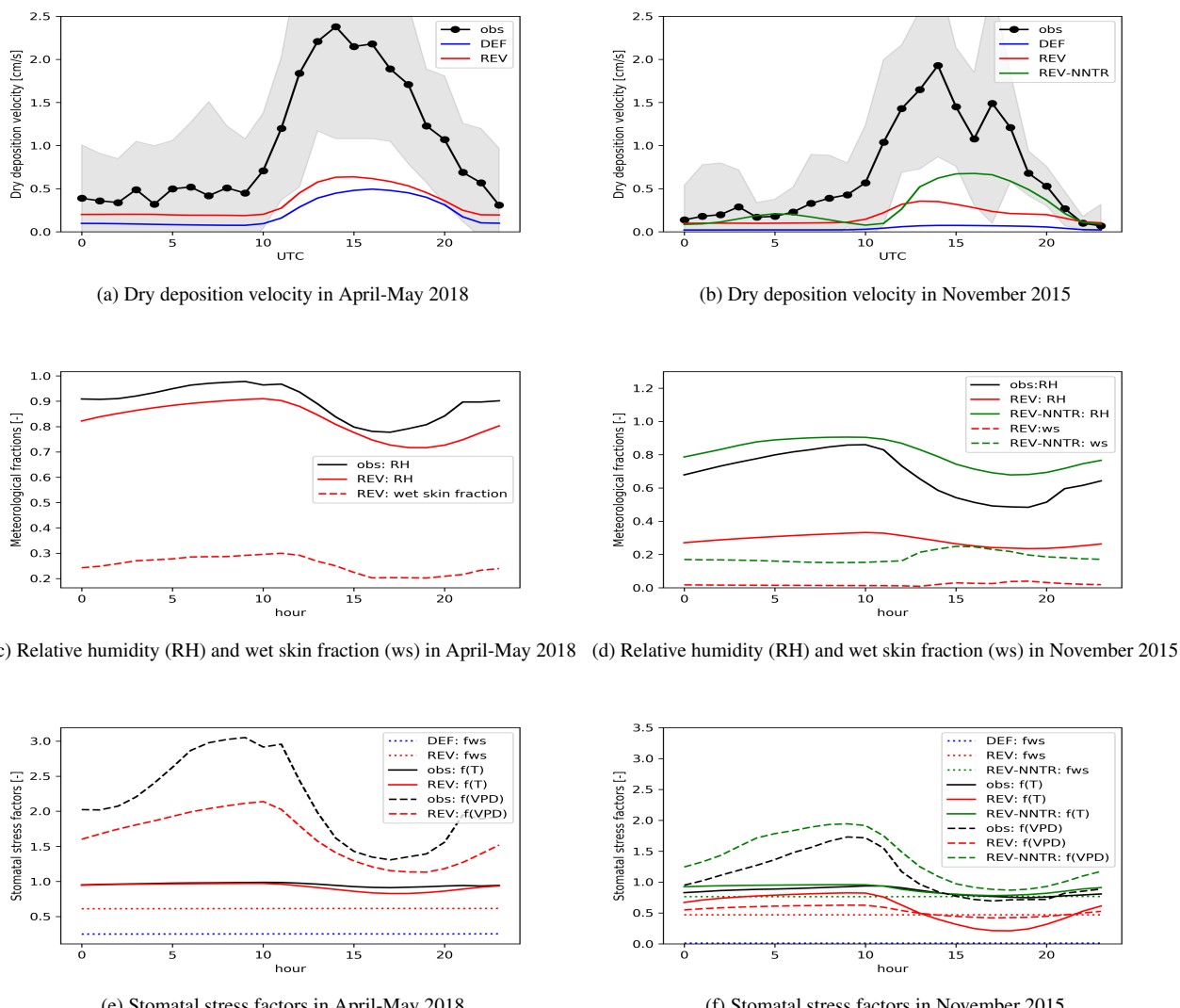

(a) Dry deposition velocity in April-May 2018

(b) Dry deposition velocity in November 2015

(c) Relative humidity (RH) and wet skin fraction (ws) in April-May 2018

(d) Relative humidity (RH) and wet skin fraction (ws) in November 2015

(e) Stomatal stress factors in April-May 2018

(f) Stomatal stress factors in November 2015

**Figure 5.** Diurnal cycles of measured (obs) and modelled (DEF, REV, REV-NNTR: free-running $f(T)$ and $f(VPD)$ for evapotranspiration) ozone dry deposition velocities in wet and dry season at ATTO (gray: standard deviation).

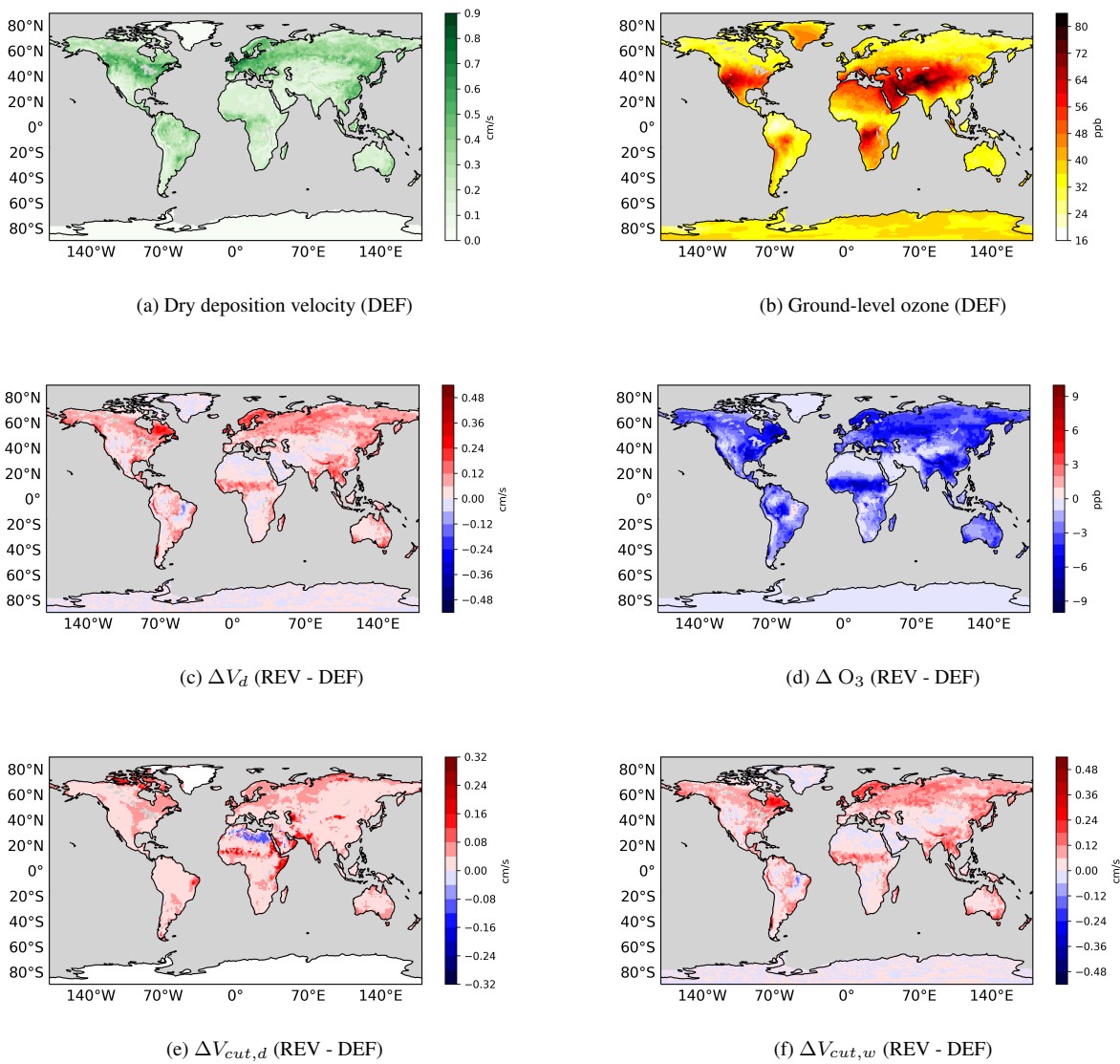

**(a) Dry deposition velocity (DEF)**

**(b) Ground-level ozone (DEF)**

**(c) $\Delta V_d$ (REV - DEF)**

**(d) $\Delta$ O$_3$ (REV - DEF)**

**(e) $\Delta V_{cut,d}$ (REV - DEF)**

**(f) $\Delta V_{cut,w}$ (REV - DEF)**

**Figure 6.** Multiyear (2010-2015) mean absolute values and changes in boreal summer: i.e, difference between revised and default scheme (REV-DEF).

stomata are stressed by temperature and VPD. This effect is overpredicted by the model, as the humidity over the Amazon forest is probably too low in the model (see Fig. A3). The stress factors are shown in Figure A4d and A4c.

However, the overall decrease in ozone concentration dampens the impact of the change in dry deposition flux. In total, the changes by the revised dry deposition scheme increase the multiyear mean (2010-2015) loss of ozone by dry deposition from 946 $\mathrm{Tg\,yr^{-1}}$ to 1001 $\mathrm{Tg\,yr^{-1}}$ (Young et al., 2018; Hu et al., 2017). Accordingly, (boreal) summer ground-level ozone

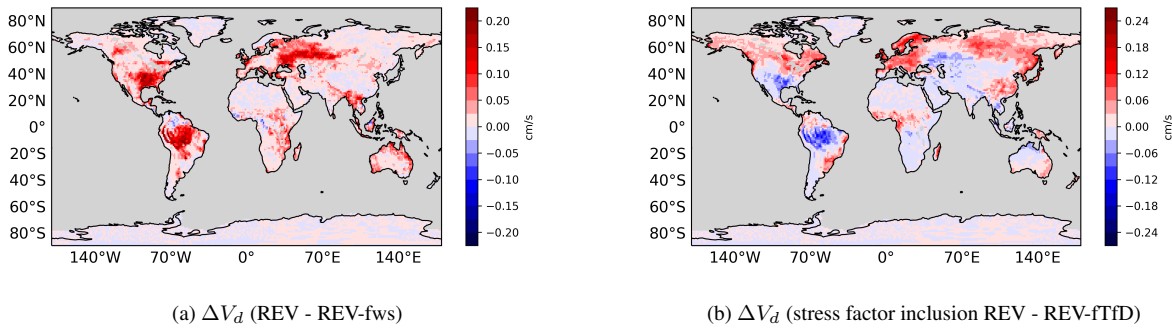

(a) $\Delta V_d$ (REV - REV-fws)

(b) $\Delta V_d$ (stress factor inclusion REV - REV-fTfD)

**Figure 7.** Mean changes (2010) of dry deposition velocity in boreal summer. (a) $f(W_s)$ modification, (b) Temperature and VPD stress

over land is reduced by up to 12 ppb (24%) peaking over Scandinavia, Asia, central Africa and East Canada (Fig. 6d). In the Northern Hemisphere, also the zonal mean of the tropospheric ozone mixing ratio show a noticeable reduction far from the ground compared to the default scheme (Fig. 9c). This has the potential to reduce the positive bias of tropospheric ozone on the Northern Hemisphere (20 %) reported by Young et al. (2018). However, besides ozone also other atmospheric tracer gases are affected by the change in dry deposition. The global annual dry deposition flux of odd oxygen $(O_x)^2$, which includes many important troposheric trace gases, increases from 978 $Tg\,yr^{-1}$ to 1032 $Tg\,yr^{-1}$ due to the revision. This is in good agreement with the reported numbers by Hu et al. (2017) and Young et al. (2018). In Fig. 8, we show additionally the absolute and relative change of the multi-year annual average dry deposition loss of $SO_2$, $NO_2$,$HNO_3$ and HCHO. As a very soluble species the loss of $SO_2$ is increased by the revised dry deposition scheme whereas the predefined low cuticular and wet skin resistance of $HNO_3$ in the old scheme were replaced with the new mechanism leading to an decrease in dry deposition. The altered loss of $NO_2$ and HCHO and other ozone precursors at ground level, especially soluble oxygenated VOCs contributes to the total change in ozone loss. $NO_2$ is deposited almost 40 % more significantly contributing to the net reduction in ozone production but is mostly counterbalanced by other processes. The change of HCHO dry deposition flux is small on a global and annual scale and only important regionally, most in (boreal) summer, when it decreases HCHO at ground level (Fig.10b) by up to 25 %. Thereby, the change in wet uptake is highest but is partially counterbalanced by other effects. This leads to lower $HO_2$-production from HCHO photooxidation and lower $NO$-to-$NO_2$ conversion and thus lower ozone production (Seinfeld and Pandis, 2016). These effects also impact the OH mixing ratio (Fig. 9b, 9d) which control the methane lifetime predicted by the model. However, for a clearer effect, a longer simulated time period would be needed. A detailed analysis of the trace gas budgets is beyond the scope of this manuscript and will be investigated in a subsequent study.

---

[2]$O_x \equiv$ O $+O_3$ $+NO_2$ $+2NO_3$ $+3N_2O_5$ $+HNO_3$ $+HNO_4$ $+BrO$ $+HOBr$ $+BrNO_2$ $+2BrNO_3$ $+PAN$

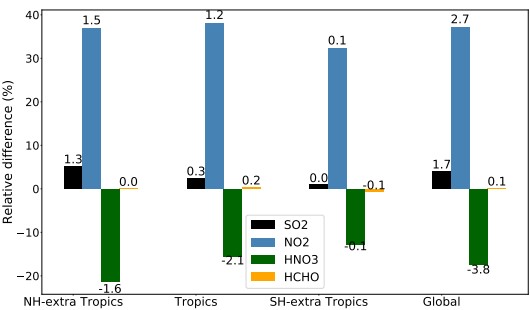

**Figure 8.** Relative change [%] and absolute change [Tg/yr] (numbers on bars) of annual global loss by dry deposition of $O_3$, $SO_2$, $HNO_3$, $HCHO$ (REV-DEF)

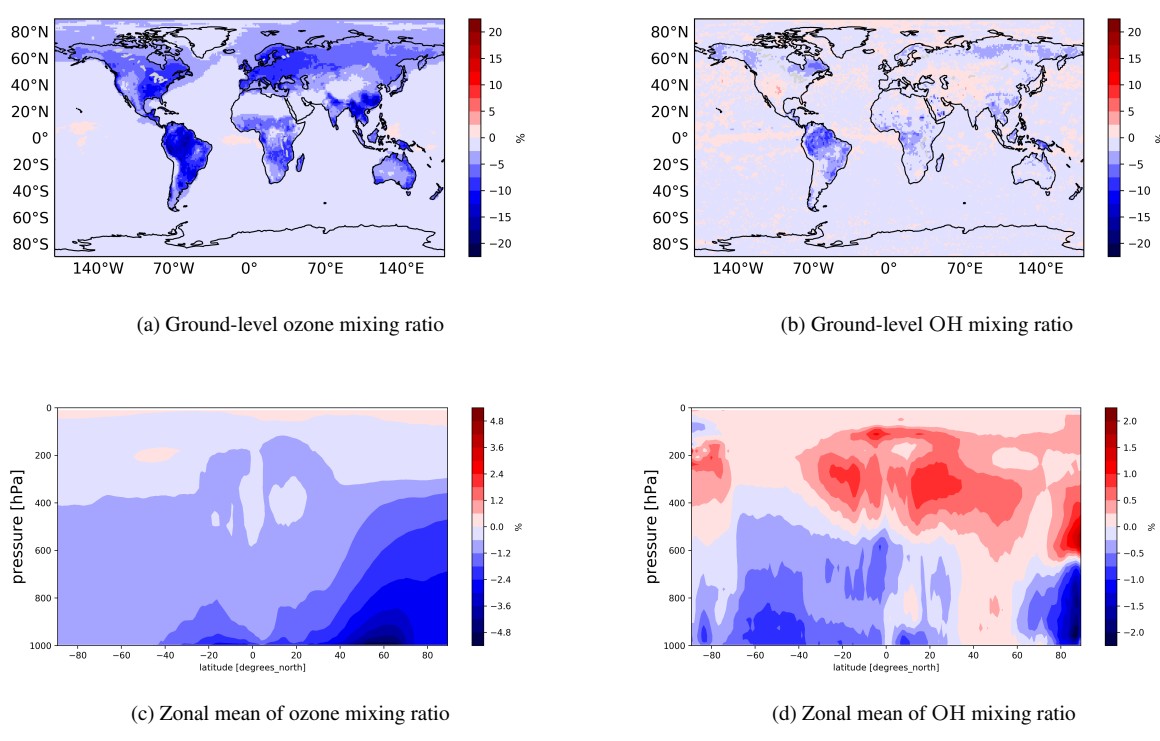

(a) Ground-level ozone mixing ratio

(b) Ground-level OH mixing ratio

(c) Zonal mean of ozone mixing ratio

(d) Zonal mean of OH mixing ratio

**Figure 9.** Relative change of multiyear (2010-2015) mean (DEF-REV)

## 6 Uncertainties in modelling stomatal conductance

Dry deposition is a highly uncertain term in modelling ozone pollution (Young et al., 2018; Clifton et al., 2020a). Its representation is general limited by a lack of measurements and process understanding but also too a large extent driven by the quality

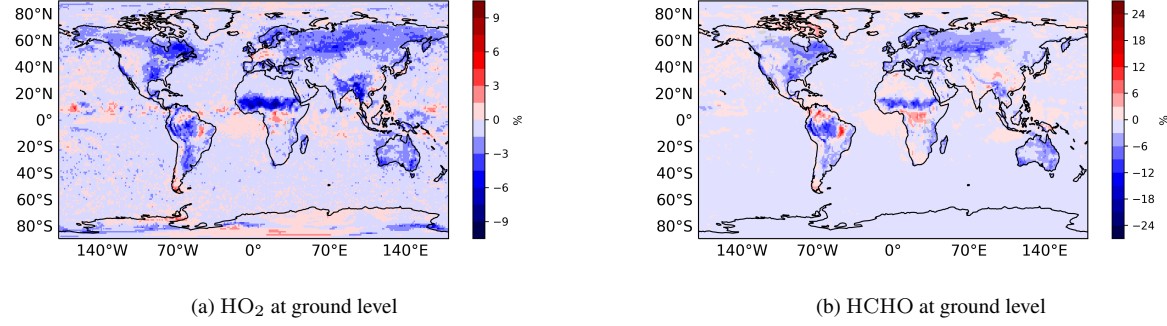

(a) HO$_2$ at ground level       (b) HCHO at ground level

**Figure 10.** Relative change of multiyear (2010-2015) boreal summer mean (DEF-REV)

of land cover information (Hardacre et al., 2015; Clifton et al., 2020b). Although the dry deposition scheme by Wesely (1989) is commonly used in global and regional models (e.g. MOZART, GEOS-Chem) the approach has some constraints (Hardacre et al., 2015). The disadvantage of the big leaf approach used in MESSy is that a vertical variation of leaf properties, affect-

ing for instance the attenuation of solar radiation is not considered (e.g., Clifton et al., 2020b). Regarding stomatal uptake, we neglect the mesophyll resistance as reactions inside the leaf are commonly assumed to not limit stomatal ozone uptake whereas, besides mostly supporting laboratory studies (e.g., Sun et al., 2016), a few contradicting findings exist (e.g., Tuzet et al., 2011). The here used empirical multiplicative algorithm by Jarvis (1976) for stomatal modelling has one general draw-back concerning that the environmental responses to stomata are treated clearly in contrast to experimental evidence (Damour

et al., 2010). However, Jarvis-type models have been shown to be able to compete with the semi-mechanistic $A_{net} - g_s$ models which link stomatal uptake to the $CO_2$ assimilation during plant photosynthesis (Fares et al., 2013; Lu, 2018). The critics in Fares et al. (2013) that the Jarvis model cannot capture the afternoon depression of ozone dry deposition is due to the original used VPD stress factor which has been replaced here by a mechanistic one based on the optimised exchange of $CO_2$ and water by plants (Katul et al., 2009). Furthermore, a larger set of land cover types is expected to improve the vegetation dependent

variation of dry deposition. The parameters used to model dry deposition of stomata, cuticle and soil are biome-dependent and using generalized ones like for the input cuticular resistance can lead to differences in dry deposition (Hoshika et al., 2018). Exemplary, discrepancies for the stomatal conductance calculated with different parameter sets are shown in Fig. 11 as summer mean of 2010. Thereby, the temperature stress factor have been calculated as in Eq. 6 using the obtained surface temperature by EMAC (Fig. 11 (a),(c)) and applied to the model (DEFAULT) stomatal conductance (Eq. 17) with two different parameter

sets for coniferous and mixed forest by Simpson et al. (2012)[3] and Zhang et al. (2003)[4]. Jarvis (1976) obtained the parameters from a set of measurements in mixed hardwood/coniferous forest in Washington. In general, the parameters are related to mea-surements where the absolute values are influenced by multiple factors like genotype and local climatic conditions (Sulis et al.,

---

[3] used parameters:$T_{min} = 0°C, T_{opt} = 18°C, T_{max} = 36°C$
[4] used parameters:$T_{min} = -3°C, T_{opt} = 21°C, T_{max} = 42°C$

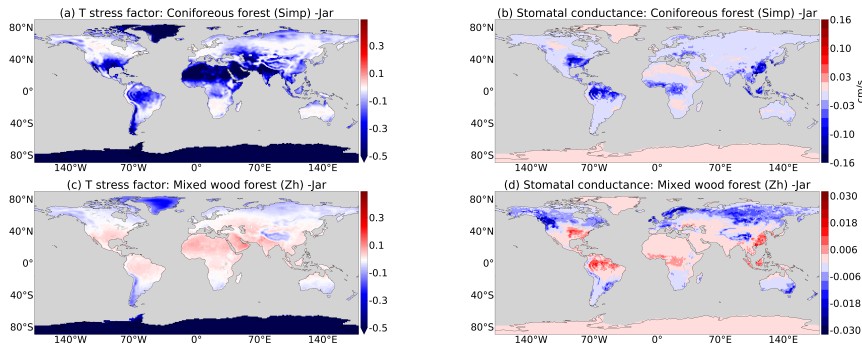

**Figure 11.** Absolute difference of stomatal conductance applied with the temperature stress factor calculated for two different parameter sets by Simpson et al. (2012) (Simp) and Zhang et al. (2003) (Zh) ein comparison with the here used parameter set by Jarvis (1976) (Jar)

2015; Tuovinen et al., 2009; Hoshika et al., 2018). So, for global modelling mostly simplified parameters have to be used like in EMEP (Simpson et al., 2012).

## 7 Sensitivity to model resolution

The simulation of dry deposition depends on meteorology including boundary layer processes, radiation (cloud distribution and reflectivity) and ozone chemistry as well as on input fields like vegetation density (LAI) (Jones, 1992). Model horizontal resolution inherently affects the amplitude and distribution of (regridded) surface processes and the artificial dilution of ozone precursors that are emitted. This aspect is investigated here by analysing simulations at three different spatial resolutions: 2.8° x2.8°, 1.9° x1.9° and 1.1° x1.1° (REST42, REST63, REV (T106) in Tab. 1). In Figure 12a the resolution dependency is shown for the annual dry deposition flux of ozone on different continental regions. The annual dry deposition fluxes differ by up to 40 $\mathrm{Tg\,yr^{-1}}$ globally between the different resolutions, with highest dry deposition at high resolution (T106). For the Northern Hemisphere (and consequently globally), this difference is driven by the higher annual mean ground-level ozone compared to the lower resolutions (Fig. 12c). However, this effect cannot be disentangled from the effect of decreased dry deposition velocity on ground-level ozone. Globally, increasing differences in $O_3$ are anti-correlated with relative humidity as shown in Figure 13b ($\rho = -0.8$). The impact of humidity on ozone chemistry is considered to be relatively weak (Jacob and Winner, 2009), but Kavassalis and Murphy (2017) showed for the U.S. that only dry deposition establishes the observed anti-correlation between ozone and relative humidity. A dominating positive correlation of the dry deposition flux with the velocity only occurs on the Southern Hemisphere extra-Tropics (SH_exT), which is highest between T63 and T106 (Fig. 13c). This can be attributed to discrepancies in stomatal deposition (Fig. 13d) driven by differences in humidity which might be caused by different moisture cycles and transpiration.

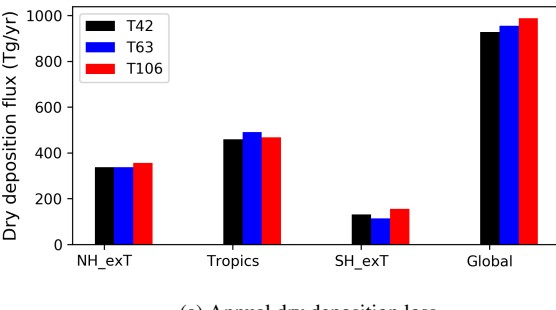

(a) Annual dry deposition loss

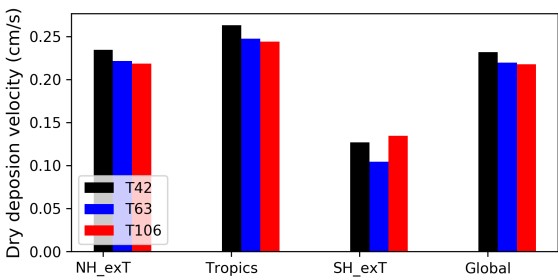

(b) Annual mean dry deposition velocity

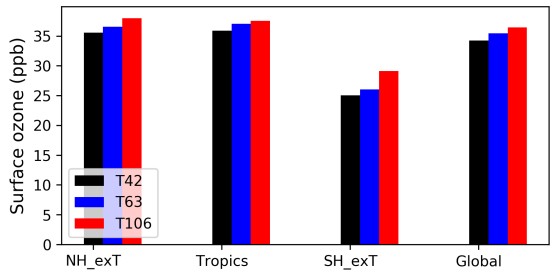

(c) Annual mean surface ozone mixing ratio

**Figure 12.** Ozone and dry deposition at three different resolutions (T42: $2.8° \times 2.8°$, T63: $1.9° \times 1.9°$, T106: $1.1° \times 1.1°$) and the different regions: Northern Hemisphere extra-Tropics (NH_exT : $90°N - 30°N$), Tropics ($30°N - 30°S$), Southern Hemisphere extra Tropics (SH_exT : $90°S - 30°S$) and the whole Earth (Global).

## 8    Conclusion and Recommendations

Dry deposition to the Earth's surface is a key process for the representation of ground-level ozone in global models. Its parametrisations constitutes a relevant part of the model uncertainty (Hardacre et al., 2015; Wu et al., 2018). Revising the
dry deposition scheme of EMAC leads to an improved representation of surface ozone in regions with a positive model ozone bias (e.g. Europe). The highest increase in ozone dry deposition is due to the implementation of cuticular uptake whose contribution is important especially during night over moist surfaces. The extension of the stomatal uptake with temperature and VPD adjustment factors accounts for the desired link of plant activity to hydroclimate as recommended by Lin et al. (2019). Especially in drought stressed regions (e.g. Citrus Orchard), the dependence on vapour pressure deficit leads to a realistic
depression of stomatal uptake at noon. Also the dependence of dry deposition on soil moisture have been modified since the current representation of soil moisture in the model is not satisfactory. Specifically, the model simulates a too dry soil for the Amazon basin causing stomatal closure and, thus an underestimation of dry deposition (Sect. 4.2). We have indications that the dry bias is a consequence of meteorological nudging in EMAC and also the missing representation of organised convection in the tropics (Mauritsen and Stevens, 2015). The sensitivity of the vegetation to droughts is comparably high in the Amazon

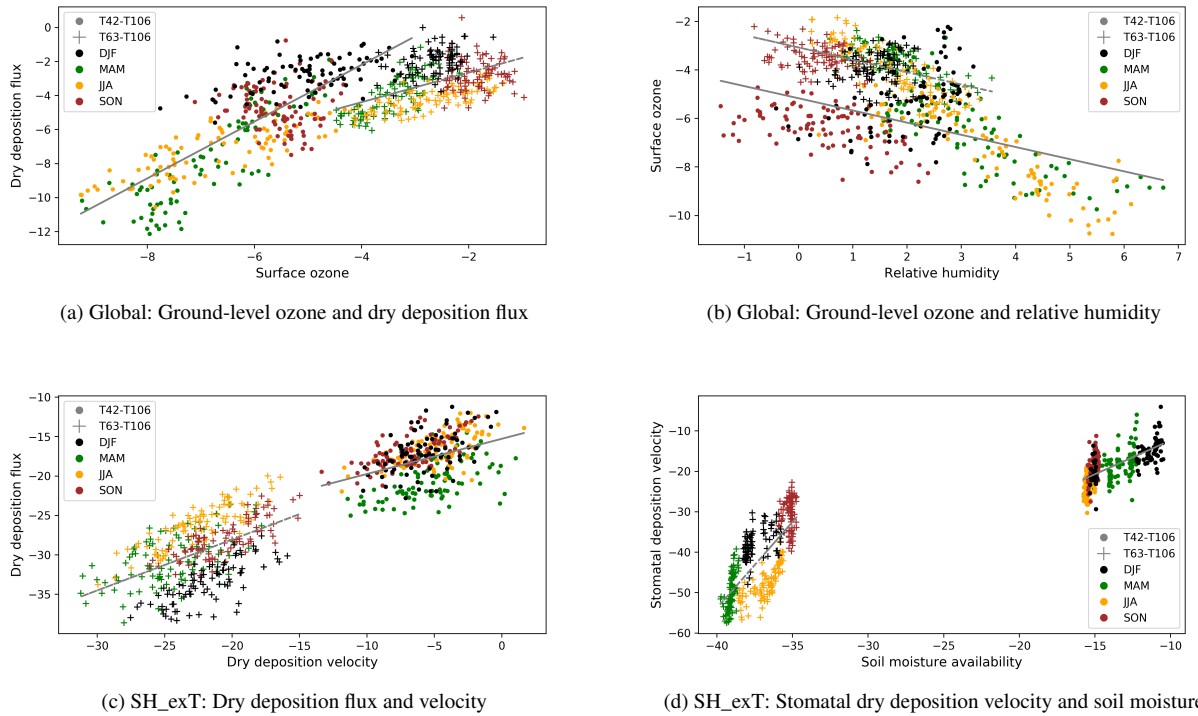

(a) Global: Ground-level ozone and dry deposition flux

(b) Global: Ground-level ozone and relative humidity

(c) SH_exT: Dry deposition flux and velocity

(d) SH_exT: Stomatal dry deposition velocity and soil moisture

**Figure 13.** Correlations of resolution dependent relative differences of ozone, dry deposition and meteorological variables for the whole Earth (global) ans the Southern Hemisphere extra-Tropics (SH_exT) for the four boreal seasons: spring (MAM), summer (JJA), autumn (SON), winter (DJF).

region because the model soil cannot hold water in the catchment for a realistic time period and exhibits a memory effect (Hagemann and Stacke, 2013). Deeper root zones or buffering of the soil moisture below the root zone would improve the water holding capacity (Hagemann and Stacke, 2013; Fisher et al., 2007). With an improved representation of soil moisture the more realistic parametrisation of the soil moisture stress on stomatal uptake could be re-enabled. In general, the inclusion of the strong link between dry deposition and meteorology reveals some limitations of the dry deposition scheme associated with the inaccurate representation of local meteorology. The results also indicate that an improved representation of important non-stomatal dry deposition like in-canopy reactions of ozone with volatile organic compounds (e.g. Citrus Orchard, Sect. 4.2) would lower the positive model-observation discrepancy. This can be achieved with the inclusion of further biogenic VOCs and an explicit parametrization of the transport dynamics in the boundary layer in model simulations (Makar et al., 2017). Explicit field measurements could foster further process understanding, which is required for a detailed process description within the models, especially over tropical rain-forests. The seasonal variability of the simulated dry deposition velocity could be further improved by using as model input the time-series of vegetation cover from an imaging products which also capture land use changes and vegetation trend that are known to impact dry deposition significantly (Wong et al., 2019).

# 9 Outlook

The representation of gaseous dry deposition in MESSy will be further improved by using the MODIS time-series of LAI
which captures multi-annual vegetation changes. As the next step of dry deposition modelling in MESSy a biome-dependent
dry deposition model coupled to $CO_2$ assimilation (White et al. 2004) will be applied. Biome-dependent vegetation cover
information, required for this scheme, are then provided by global input data which, however, represent only the annual cycle
of vegetation. Coupling MESSy to the recently available dynamic vegetation model LPJ-GUESS providing detailed vegetation
information with the temporal variability required for a climate model could be a further improvement. By now the one-way
coupling of LPJ-GUESS as a MESSy submodel is only in the initial evaluation phase of the coupling with the atmospheric
model (Forrest et al., 2020).

*Data availability.* The measurement data at Ontario is freely available at http://data.ec.gc.ca/data/air/monitor/special-studies-of-atmospheric-
gases-particles-and-precipitation-chemistry/borden-forest-ozone-and-sulphur-dioxide-dry-deposition-study with the 'Open Gouvernment Licence-
Canada' (https://open.canada.ca/en/open-government-licence-canada). The measurement data at Hytiälä (Creative Commons 4.0 Attribution
(CC BY) license https://creativecommons.org/licenses/by/4.0/) can be accessed at https://avaa.tdata.fi/web/smart/smear/download. The data
from Lindcove station (Fares) were provided by S. Fares (Fares et al., 2012). The dry deposition measurement data at Amazonian Tall Tower
Observatory was provided by Matthias Sörgel and is available on request. The used global climate reanalysis ERA5 by ECMWF are available
through the Climate Data Store (https://cds.climate.copernicus.eu).

*Code availability.* The Modular Earth Submodel System (MESSy) is continuously further developed and applied by a consortium of insti-
tutions. The usage of MESSy and access to the source code is licenced to all affiliates of institutions which are members of the MESSy
Consortium. Institutions can become a member of the MESSy Consortium by signing the MESSy Memorandum of Understanding. More
information can be found on the MESSy Consortium Website http://www.messy-interface.org. The code presented here has been based on
MESSy version 2.54 and will be available in the next official release (version 2.55). The exact code version used to produce the results of
this paper is archived in the MESSy code repository and can be made available to members of the MESSy community upon request.

## Appendix A: Default dry deposition scheme

The default dry deposition scheme of MESSy uses the following equations described in Kerkweg et al. (2006).

Surface resistance over vegetation (in $\mathrm{s\,m^{-1}}$):

$$\frac{1}{R_{s,veg}(X)} = \frac{1}{R_{can} + R_{s,soil}(X) + R_{qbr,veg}(X)} + \frac{LAI}{r_{cut}(X)} + \frac{LAI}{r_{stom,corr}(X) + r_{mes}(X)} \tag{A1}$$

where $R_{can}(X)$, $R_{s,soil}(X)$, $R_{qbr,veg}(X)$ are the in-canopy aerodynamic resistance, the soil resistance and the quasi-laminar boundary resistance at canopy scale (in $\mathrm{s\,m^{-1}}$). $r_{cut}(X)$, $r_{stom,corr}(X)$ and $r_{mes}(X)$ are the cuticular resistance, stomatal resistance and mesophyll resistance at leaf scale scaled with Leaf Area Index (LAI in $\mathrm{m^2\,m^{-2}}$) to canopy scale.

Stomatal resistance:

$$r_{stom,corr} = \frac{r_{stom}(PAR)}{fws} \cdot \frac{D_{\mathrm{H_2O}}}{D(\mathrm{O_3})} \tag{A2}$$

Soil moisture stress function:

$$f(W_s) = \begin{cases} 1 & W_s(t) \geq W_{cr}(=75\%) \\ \frac{W_s(t) - W_{pwp}}{W_{cr} - W_{pwp}} & W_{pwp} < W_s(t) < W_{cr} \\ 0 & W_s(t) \leq W_{pwp}(=35\%) \end{cases} \tag{A3}$$

Cuticular resistance:

$$r_{cut}(X) = \frac{r_{cut}(\mathrm{O_3})}{10^{-5} \cdot H(\mathrm{O_3}) + s_{reac}(\mathrm{O_3})} \tag{A4}$$

where $r_{cut}\mathrm{O_3} = 1e-5\,\mathrm{s\,m^{-1}}$, $H(\mathrm{O_3}) = 0.01$ and $s_{reac} = 1$.

Wet skin resistance:

$$R_{ws}(\mathrm{O_3}) = \left[ \frac{1/3}{R_{ws}(\mathrm{SO_2})} + 10^{-7} \cdot H(\mathrm{O_3}) + \frac{s_{reac}(\mathrm{O_3})}{R_{cut,w}(\mathrm{O_3})} \right]^{-1} \tag{A5}$$

where $R_{ws}(\mathrm{O_3}) = 2000\,\mathrm{s\,m^{-1}}$ and $R_{ws}(\mathrm{SO_2}) = 100\,\mathrm{s\,m^{-1}}$.

## Appendix B: Evapotranspiration

Plants play a key role in the water and energy cycle and thus contribute to the land-atmosphere coupling, which drives the global climate. In this context, transpiration is an important process, as plants loose water during the necessary $CO_2$ uptake via their stomata. The amount depends on the aperture behaviour of the respective plant in the respective environmental conditions (Katul et al., 2012). Thus, the latent heat flux incorporates the canopy resistance. The formulation is based on the Monin-Obukov stability theory:

$$E = \rho C_h |\boldsymbol{v}| \beta (q_a - h q_s(T_s, p_s)) \qquad \beta = \left[ 1 + \frac{C_h |\boldsymbol{v}| R_{stom}}{fws} \right]^{-1} \tag{B1}$$

where $\rho$ is the density of air, $|\boldsymbol{v}|$ is the absolute value of the horizontal wind speed, $C_h$ is the transfer coefficent of heat whereas $r_a = 1/(C_h|\boldsymbol{v}|)$. $q_s$ and $q_a$ are the saturation-specific humidity and the atmospheric specific humidity whereas the relative humidity h at the surface limits the evapotranspiration from bare soil. $\beta$ determines the ratio of transpiration between water stressed plants ($\beta <1$) and well-watered plants ($\beta =1$) (Giorgetta et al., 2013; Schulz et al., 2001). The formular for the canopy stomatal resistance $R_{stom}$ is given in Eq. 5. In order to adapt the transpiration to temperature and vapour pressure deficit the T and VPD adjustment factors can be applied to $R_{stom}$ inversely like in the new dry deposition scheme via $izwet = 1$ in the VERTEX $\&CTRL$ namelist. The modification of the soil moisture stress function $f(W_s)$ (old: Eq. A, new: Eq. 12) affects evapotranspiration directly.

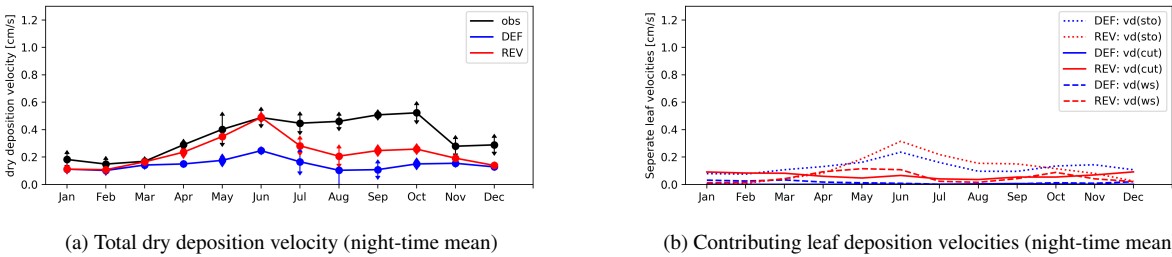

(a) Total dry deposition velocity (night-time mean)

(b) Contributing leaf deposition velocities (night-time mean)

**Figure A1.** Measured and modelled (DEF, REV) annual cycle at Borden forest

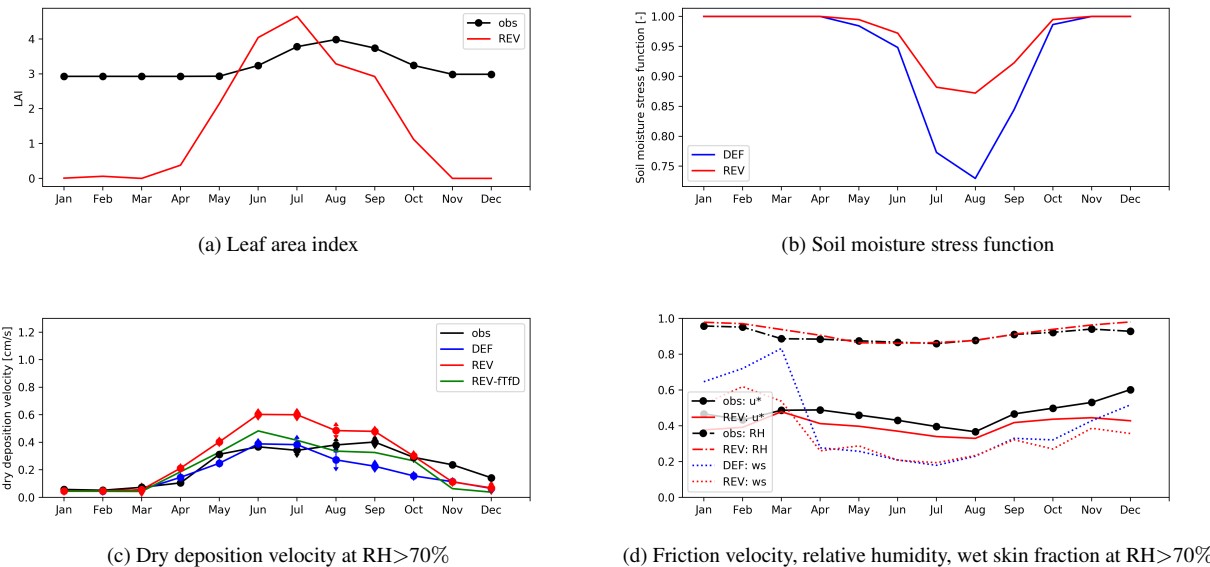

(a) Leaf area index

(b) Soil moisture stress function

(c) Dry deposition velocity at RH>70%

(d) Friction velocity, relative humidity, wet skin fraction at RH>70%

**Figure A2.** Measured (obs) and modelled (DEF, REV) multiyear (2010-2012) and REV-fTfD (2010) annual cycle at Hyytiälä

*Author contributions.* D.T. (and A.K.) initiated and supervised the study. D.T. and T.E. discussed the model developments which were implemented by A.K. and T.E.. H.O. originally wrote the MESSy vertical diffusion submodel VERTEX. S.F. provided the measurement data from Lindcove and further related theoretical calculations. I.M. did the dry deposition measurements at Hytiälä and gave related support. T.E. performed the EMAC simulations, the data analyses, prepared the figures and wrote the manuscript.

*Competing interests.* The authors declare no competing financial interests.

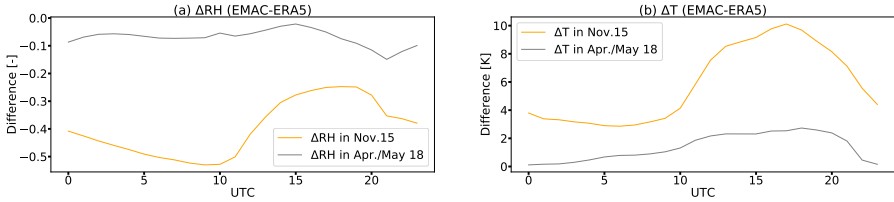

**Figure A3.** Differences of meteorology between EMAC and ERA5 at ATTO

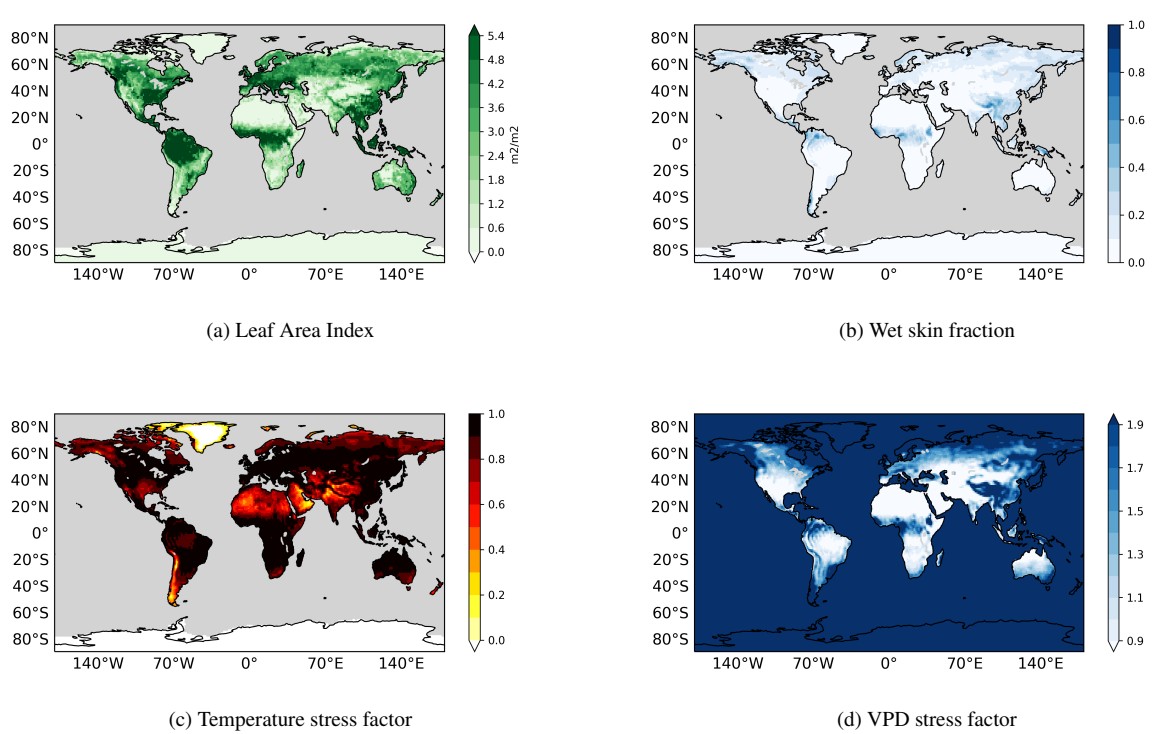

(a) Leaf Area Index

(b) Wet skin fraction

(c) Temperature stress factor

(d) VPD stress factor

**Figure A4.** Boreal summer mean vegetation and meteorological variables predicted by EMAC

*Acknowledgements.* The work described in this paper has received funding from the Initiative and Networking Fund of the Helmholtz Association through the project "Advanced Earth System Modelling Capacity (ESM)". The content of this paper is the sole responsibility of the author(s) and it does not represent the opinion of the Helmholtz Association, and the Helmholtz Association is not responsible for any use that might be made of the information contained. The author(s) acknowledge the Environment and Climate Change Canada and the United States Environmental Protection Agency for the provision of the dry deposition velocity data at the Borden forest measurement station. Moreover, the personnel at SMEAR II station of INAR – Institute for Atmospheric and Earth System Research, University of Helsinki, Finland, is acknowledged. Concerning the measurement data from Amazonian Tall Tower, we thank the Instituto Nacional de Pesquisas da Amazonia (INPA) and the Max Planck Society for continuous support. We thank for the support by the German Federal Ministry of Education

and Research (BMBF contracts 01LB1001A, 01LK1602B and 01LP1606B) and the Brazilian Ministério da Ciência, Tecnologia e Inovação (MCTI/FINEP contract 01.11.01248.00) as well as the Amazon State University (UEA), FAPEAM, LBA/INPA and SDS/CEUC/RDS-Uatumã. The measurements were conducted by Matthias Sörgel, Anywhere Tsokankunku, Stefan Wolff and Rodrigo Souza. For the usage of data from the ERA5 global climate reanalysis (Generated using Copernicus Atmosphere Monitoring Service Information [2020]) we acknowledge the Copernicus Climate Change and Atmosphere Monitoring Service (https://apps.ecmwf.int/datasets/licences/copernicus/). Neither the European Commission nor ECMWF is responsible for any use that may be made of the Copernicus information or data it contains.

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
