# Peer review of "A revised dry deposition scheme for land-atmosphere exchange of trace gases in ECHAM/MESSy v2.54"

_Geoscientific Model Development, 2020_

## Referee Comment (RC1) · Dennis Baldocchi (Referee) · 20 Jul 2020

A revised dry deposition scheme for land-atmosphere exchange of trace gases in ECHAM/MESSy v2.54 Tamara Emmerichs1, Astrid Kerkweg1, Huug Ouwersloot2, Silvano Fares3, Ivan Mammarella4, and Domenico Taraborrelli

The field of dry deposition has had periods of ups and downs in activity and research. Unfortunately algorithms in important models have been fossilized to consider the Wesely model of 1989. While that was a very good and appropriate algorithm 30 years ago, we know more about land surface fluxes, how to model stomatal conductance and have been datasets and parameterization information in 2020. So, I was excited to see

this paper.

I see the main contributions are

The default dry deposition scheme has been extended with adjustment factors to predict stomatal responses to temperature and vapour pressure deficit. Furthermore, an explicit formulation of the non-stomatal deposition to the leaf surface (cuticle) dependent on humidity has been implemented based on established schemes. Finally, the soil moisture availability function for plants has been revised to be consistent with the simple hydrological model available in EMAC.

The authors make a good case for this work and its significance as 'the revision of the process parameterisation as documented here has the potential to significantly reduce the overestimation of tropospheric ozone in global models'.

This paper is a steps in the right direction, but revolves around the over parameterized Jarvis stomatal model that was used in the 80s with more adjustment factors. Many of us, including Piers Sellers, have abandoned the Jarvis model in land-surface modeling of water and carbon fluxes because it lead to stomatal suicide. Others have adopted the Ball-Berry approach, with better fidelity

Baldocchi, D. D., and T. Meyers (1998), On using eco-physiological, micrometeorological and biogeochemical theory to evaluate carbon dioxide, water vapor and trace gas fluxes over vegetation: a perspective, Agricultural and Forest Meteorology, 90(1-2), 1-25.

I don't view this 'new'model as an improvement by going back to the Jarvis model for stomatal conductance. There has been many advances in stomatal modeling worth considering in 2020.

Wang, Yujie, John S. Sperry, William RL Anderegg, Martin D. Venturas, and Anna T. Trugman. "A theoretical and empirical assessment of stomatal optimization modeling." New Phytologist (2020).

Medlyn, B. E., Duursma, R. A., Eamus, D., Ellsworth, D. S., Prentice, I. C., Barton, C. V., ... & Wingate, L. (2011). Reconciling the optimal and empirical approaches to modelling stomatal conductance. Global Change Biology, 17(6), 2134-2144.

Personally, I'd like to see some connection with ecosystem photosynthesis scaling with stomatal conductance. There has been excellent advances modeling both that could be coupled with a stomatal and dry deposition model, for instance.

Jiang, C., and Y. Ryu (2016), Multi-scale evaluation of global gross primary productivity and evapotranspiration products derived from Breathing Earth System Simulator (BESS), Remote Sensing of Environment, 186, 528-547, doi:http://dx.doi.org/10.1016/j.rse.2016.08.030.

De Kauwe, Martin G., et al. "A test of an optimal stomatal conductance scheme within the CABLE land surface model." Geoscientific Model Development (2015): 431-452.

In writing the introduction, there has been some recent workshops on dry deposition, newer long term studies and a very good review that should be cited and considered

Clifton, O. E., Fiore, A. M., Massman, W. J., Baublitz, C. B., Coyle, M., Emberson, L., ... & Guenther, A. B. (2020). Dry deposition of ozone over land: processes, measurement, and modeling. Reviews of Geophysics, 58(1), e2019RG000670.

Clifton, O. E., A. M. Fiore, J. W. Munger, S. Malyshev, L. W. Horowitz, E. Shevliakova, F. Paulot, L. T. Murray, and K. L. Griffin (2017), Interannual variability in ozone removal by a temperate deciduous forest, Geophysical Research Letters, 44(1), 542-552, doi:10.1002/2016gl070923.

Clifton, O. E., Paulot, F., Fiore, A. M., Horowitz, L. W., Correa, G., Baublitz, C. B., ... & Hogg, A. J. (2020). Influence of dynamic ozone dry deposition on ozone pollution. Journal of Geophysical Research: Atmospheres, 125(8), e2020JD032398.

I am of mixed feelings of this work. I find the model algorithm dated and not an improvement. On the other hand there has been a dearth of long term flux measurements and

use of those data to test the performance of a model, as it done here.

To my opinion this would be much better paper by using modern, better state of art stomatal models that couple carbon and water fluxes and test the performance against a year of flux measurements. Then I would feel the work is new, novel and a significant improvement over the past work.

I also like the use of 4 contrasting flux datasets. This too is an advance in model testing.

For example regarding performance, we learn

'As seen from the comparison of stomatal resistance values (Fig. 4d) the model underestimates the stomatal uptake. This is because the irrigation of the Orchard leads to cooling sustained evapotranspiration and keeps f(T) low.Thus in the model, a too high temperature stress act on the stomata'

My alternative hypothesis is that this bias may disappear with a coupled carbon-water stomatal conductance model.

If I have learned anything over my career it is the power and importance of multiple constraints. Sadly, the Jarvis model does not deliver. It was great circa 1976 and helped us think about the role of stomata on dry deposition in the 1980s, but that is its extent of being good enough.

Fig 3 would be better if error bars were added, given these are monthly means.

I do like the global upscaling. It helps address the 'so what?' question and does produce some multiple constraint with regards to getting pollution right, as we see in Fig. 6.

My bottom line is that this paper can be remedied. It has lots of strengths worth keeping. And the spirit of the work is good.

Regarding conclusion

The seasonal variability of the simulated dry deposition velocity could be further improved by using as model input the time-series of vegetation cover from an imaging products which also capture land use changes and vegetation trend that are known to impact dry deposition significantly.

Connection to phenology modeling or observation is key to getting the seasonality in LAI correct and the fluxes right. So Yes this is an important aspect of the model. I'd like to see it in the 'new model'.

If the model had already coupled water and carbon phenology should be part of it.

---

## Author Comment (AC1) · 5 Aug 2020

**Answer to referee comments GMD**

July 2020

**Comment by D. Baldocchi**

Dear referee, many thanks for your review. Here are our replies:

> *The field of dry deposition has had periods of ups and downs in activity and research. Unfortunately algorithms in important models have been fossilised to consider the Wesely model of 1989. While that was a very good and appropriate algorithm 30 years ago, we know more about land surface fluxes, how to model stomatal conductance and have been datasets and parameterization information in 2020. So, I was excited to see this paper. I see the main contributions are The default dry deposition scheme has been extended with adjustment factors to predictstomatal responses to temperature and vapour pressure deficit. Furthermore, an explicit formulation of the non-stomatal deposition to the leaf surface (cuticle) dependent on humidity has been implemented based on established schemes. Finally, the soil moisture availability function for plants has been revised to be consistent with the simple hydrological model available in EMAC. The authors make a good case for this work and its significance as 'the revision of the process parameterisation as documented here has the potential to significantly reduce the overestimation of tropospheric ozone in global models'.*

Reply: The article documents a revision of the existing dry deposition scheme in EMAC not a complete new implementation. The idea is to improve the existing scheme based on the already available information in the model (i.e. without detailed phenology information etc.) because model results show that a more precise representation could lower the overestimation of ozone by models. With the current model version, these developments can only draw on limited vegetation information without details on cover and phenology. Dry deposition of trace gases is represented by the "resistance-in-series" scheme of Wesely (1989). The stomatal uptake was firstly only based on the response to incoming solar radiation developed by Sellers (1985) which is known to be an important fluctuation factor (Dawson et al. 2010), and a soil moisture stress factor. The further developments were build on this common dry deposition scheme. For the extension with additional stress factors, we adopt the multiplicative principle and the

temperature stress factor by Jarvis (1976). This principle is commonly used in second-generation LSM schemes due to its computational efficiency, adaptability and simplicity (Pitman et al. 2003, Clifton et al. 2020) and has been shown to capture 95 % of the observed variability of stomatal conductance (Dawson et al 2010). The stomatal sensitivity to vapour pressure deficit is calculated according to the optimisation framework by Katul et al. (2009) which maximises the use of carbon under a minimal cost of water inside the plant. This concept accounts for the water cost of carbon without specifying the stomatal response to VPD and $CO_2$ in advance and agrees well with experimental data (Katul et al. 2009). Hence, by adding also the stomatal response to temperature and vapour pressure deficit within this study, the key responses of stomates are represented (Pitman et al. 2003).

> *This paper is a steps in the right direction, but revolves around the over parameterized Jarvis stomatal model that was used in the 80s with more adjustment factors. Many of us, including Piers Sellers, have abandoned the Jarvis model in land-surface modeling of water and carbon fluxes because it lead to stomatal suicide. Others have adopted the Ball-Berry approach, with better fidelity.*

Reply: Comparing to measurement data, several studies found that Jarvis-type models can compete with Ball-Berry models in explaining observed stomatal conductance and stomatal ozone flux to vegetation (Hoshika et al. 2017, Ran et al. 2017) whereas both have different limitations and advantages (Lu 2018, Farquhar et al. 1980). The performance of both models depend certainly on the choice of parameters (Sulis et al. 2015, Lu 2018). The mentioned "stomatal suicide" as major critique to the Jarvis model has been experienced in EMAC and is attributed to the lack of soil moisture storage in some regions. It is solved currently by adapting the soil moisture stress factor to the used soil representation. Moreover, the stress factor dependent on VPD (Katul et al. 2009), that we use, exerts a stronger control on evapostranspiration that the original factor proposed by Jarvis. For comparison, at VPD = 5 kPa stomatal conductance is predicted to decrease by about 50% and $< 10\%$ according to Katul et al. (2009) and Jarvis (1976), respectvely. A further amelioration of the EMAC model dry bias in the Amazon is brought by the use of VPD factor by Katul et al. (2009) only in simulations without meteorological nudging (not shown in the manuscript). The usage of the Ball-Berry approach is constrained by the availability of detailed information on plant microphysics which determine the parameters. Due to the current limitations of EMAC in this regard, described above, an implementation would build on many assumptions concerning the representation at global scale.

> *"I don't view this 'new' model as an improvement by going back to the Jarvis model for stomatal conductance. There has been many advances in stomatal modelling worth considering in 2020."*

Reply: With regard to the developments of stomatal conductance models in the last years the approach used here is dated but in EMAC this represents a

significant improvement compared to the existing parametrization.The adaptability, simplicity and computer efficiency makes it attractive for the use at global scale and the usage of parametrizations for radiation response and VPD stress are different from the one used in Jarvis (1976).

> *Personally, I'd like to see some connection with ecosystem photosynthesis scaling with stomatal conductance. There has been excellent advances modeling both that could be coupled with a stomatal and dry deposition model, for instance.*

Reply: We agree with the Referee but unfortunately these developments are limited by the minimal ecosystem representation in the EMAC model. Implementing a mechanistic approach which connect stomatal conductance to plant photosynthesis is definitely intended for EMAC once a vegetation model with the sufficient details and well-constrained parameters will be available.

> *In writing the introduction, there has been some recent workshops on dry deposition, newer long term studies and a very good review that should be cited and considered.*

Reply: We will add a paragraph on the current research status of dry deposition to the introduction considering this studies .

> *I am of mixed feelings of this work. I find the model algorithm dated and not an improvement. On the other hand there has been a dearth of long term flux measurements and use of those data to test the performance of a model, as it done here. To my opinion this would be much better paper by using modern, better state of art stomatal models that couple carbon and water fluxes and test the performance against a year of flux measurements. Then I would feel the work is new, novel and a significant improvement over the past work.*

Reply: With regard to the mentioned limitations and the current status of the dry deposition parametrization in EMAC, our development can be seen as an intermediate stage on the way to a "state-of-the-art" dry deposition scheme. For the stomatal part, major dependencies to meteorology have been established whereas the implementation of the cuticular pathway contributes to a global enhancement of dry deposition especially of soluble organic species that are ozone precursors. Furthermore, the study has a significance for the MESSy community as first technical description and evaluation of the vertical exchange submodel VERTEX.

> *I also like the use of 4 contrasting flux datasets. This too is an advance in model testing. For example regarding performance, we learn 'As seen from the comparison of stomatal resistance values (Fig. 4d) the model underestimates the stomatal uptake. This*

*is because the irrigation of the Orchard leads to cooling sustained evapotranspiration and keeps f(T) low.Thus in the model, a too high temperature stress act on the stomata'. My alternative hypothesis is that this bias may disappear with a coupled carbon-water stomatal conductance model.*

Reply: Concerning the model evaluation at Citrus Orchard, we cannot exclude that such a model might remove the bias. However, if it did, it would do it for the wrong reasons. The absence of soil water stress at Citrus Orchard (due to irrigation) is artificial and not represented in the global model. Thus, the site cannot be representative for the mostly non irrigated 1.1°x1.1° grid box including Citrus Orchard. In fact, removal of the water stress from the model greatly reduces the model bias at Citrus Orchard (see Fig. 4d).

*If I have learned anything over my career it is the power and importance of multiple constraints. Sadly, the Jarvis model does not deliver. It was great circa 1976 and helped us think about the role of stomata on dry deposition in the 1980s, but that is its extent of being good enough.*

Reply: We are aware of the limitations of the implemented model parametrization. But regarding that the developments for a global model which has only a minimal ecosystem representation available, we see the current implementation as the best achievable in EMAC without having to embark on the coupling with a dynamic vegetation model that would provide the desired constraints.

*Fig 3 would be better if error bars were added, given these are monthly means.*

Reply: Error bars can be added for all sub figures.

*I do like the global upscaling. It helps address the 'so what?' question and does produce some multiple constraint with regards to getting pollution right, as we see in Fig. 6.*

Reply: Thank you for mentioning this aspect which addresses the actual motivation of this model study. EMAC is an Atmospheric Chemistry Model which explicit chemistry and misses on the other hand details for e.g. the vegetation representation.

*My bottom line is that this paper can be remedied. It has lots of strengths worth keeping. And the spirit of the work is good.*

Reply: Regarding all the arguments mentioned above we can not be sure that implementing a simple 'Anet-$g_s$' stomatal approach relying on the scanty vegetation information available in the model could improve the representation of dry deposition in EMAC.

*"The seasonal variability of the simulated dry deposition velocity could be further improved by using as model input the time-series of vegetation cover from an imaging products which also capture land use changes and vegetation trend that are known to impact dry deposition significantly. Connection to phenology modelling or observation is key to getting the seasonality in LAI correct and the fluxes right. So Yes this is an important aspect of the model. I'd like to see it in the 'new model'. If the model had already coupled water and carbon phenology should be part of it."*

Reply: The usage of the time-series of vegetation cover from the Moderate Resolution Imaging Spectroradiometer (MODIS) is in preparation as one of the few available means to represent ecosystem phenology in the current model. However, so far only LAI data from MODIS is available in the model and remaining data like canopy height still have to be acquired. Water and carbon phenology is unfortunately not yet part of the model and will be added as part of a future planned vegetation model for EMAC.

**References**

Yen-Sen Lu, Propagation of land surface model uncertainties in simulated terrestrial system states. 2018. PhD Thesis. Universitäts-und Landesbibliothek Bonn. http://hss.ulb.uni-bonn.de/2018/5071/5071.htm

Hoshika, Y.; Fares, S.; Savi, F.; Gruening, C.; Goded, I.; De Marco, A.; Sicard, P. Paoletti, E. Stomatal conductance models for ozone risk assessment at canopy level in two Mediterranean evergreen forests. Agricultural and Forest Meteorology (2017), 234, 212-221.

Damour, Gaëlle, et al. "An overview of models of stomatal conductance at the leaf level." Plant, Cell Environment 33.9 (2010): 1419-1438.

Farquhar, Graham D., S. von von Caemmerer, and J. A. Berry. "A biochemical model of photosynthetic $CO_2$ assimilation in leaves of $C_3$ species." Planta 149.1 (1980): 78-90.

Katul, Gabriel G., Sari Palmroth, and R. A. M. Oren. "Leaf stomatal responses to vapour pressure deficit under current and CO2-enriched atmosphere explained by the economics of gas exchange." Plant, Cell Environment 32.8 (2009): 968-979.

Pitman, A. J. The evolution of, and revolution in, land surface schemes designed for climate models Int. J. Climatol., John Wiley Sons, Ltd., 2003, 23, 479-510

Sulis, M.; Langensiepen, M.; Shrestha, P.; Schickling, A.; Simmer, C. Kollet, S. J. Evaluating the Influence of Plant-Specific Physiological Parameterizations on the Partitioning of Land Surface Energy Fluxes J. Hydrometeor, American Meteorological Society, 2015, 16, 517-533

Jarvis, P. G. "The interpretation of the variations in leaf water potential and stomatal conductance found in canopies in the field." Philosophical Transactions of the Royal Society of London. B, Biological Sciences 273.927 (1976): 593-610.

Wesely, Ml. "Parameterization of surface resistances to gaseous dry deposition in regional-scale numerical models." Atmospheric environment 23.6 (1989): 1293-1304.

Sellers, Piers J. "Canopy reflectance, photosynthesis and transpiration." International journal of remote sensing 6.8 (1985): 1335-1372.

---

## Referee Comment (RC2) · Anonymous Referee #2 · 20 Sep 2020

**General Comments:**

Even there is always a severe lack of direct flux measurement, the sporadic efforts over the past 20 years still reveals a lot of new and interesting environment dependence and inter-site variabilities of gaseous dry deposition after the proposal of the ever-popular (Hardacre et al., 2015) Wesely scheme (Wesely, 1989) and its slight variants (e.g. Wang et al., 1998). Meanwhile, enormous advance has been made over modelling carbon-water exchange, and therefore stomatal modelling. And given that dry deposition has been shown as one of the major uncertainty of modelling surface ozone (Wong et al., 2019), therefore, I largely agree with the position of the first reviewer, that the effort of updating gaseous dry deposition schemes shall be welcomed and encouraged.

Yet, I doubt whether this paper is doing a good enough job in "updating" the dry deposition scheme, particularly in terms of modelling canopy resistance. Given the functional diversity of plants on the Earth, I find one of the biggest weakness of the scheme presented in this paper is the lack of biome-dependence of both its stomatal and cuticular parameters, especially given that previous works have already addressed this issue (e.g. Emberson et al., 2013; Simpson et al., 2012; Zhang et al., 2003). There is also notable weakness in evaluation of the proposed scheme, but it is much easier to address.

**Specific Comments**

Starting from stomatal conductance. I agree with the authors, that the simplicity and effectiveness of Jarvis-type parameterizations have its place in atmospheric modelling. Yet this particular ecophysiological theory itself (Jarvis, 1976) only states that stomatal conductance has multiple simultaneous constraint (mathematically, $g_s = g_{max} \prod_{i}^{n_{constraints}} f_i(X_i)$, $0 \leq f_i \leq 1$), but does not explicitly gives universal functional forms (i.e. the mathematical forms of $f_i$) and parameters of all biomes over the world. It has been explicitly shown that improperly parameterized Jarvis-type model can lead to substantial bias (Fares et al., 2013).

Earlier works of updating dry deposition schemes with Jarvis-type stomatal sub-models (e.g. Simpson et al., 2012; Zhang et al., 2003) had already been assigning stomatal parameters to each individual biome. Though one may argue that they are neither backed empirically (improperly parameterized), they are probably still working better, especially for global modelling, than one single set of stomatal parameters over all biomes. For example, Hoshika et al. (2018) empirically derive that $g_{smax}$ (maximum stomatal conductance) can vary almost ten-folds across all biomes, and the optimal temperature of stomatal opening ($T_{opt}$) generally increases as the mean annual air temperature. The Zhang and EMEP parameterizations stated above are able to qualitatively capture some features showing in Hoshika et al. (2018) (e.g. higher $g_{smax}$ for broadleaf trees and crops than boreal forests, higher $T_{opt}$ for tropical than boreal biomes), giving them more creditability when applied regionally and globally, which cannot be achieved by one single set of stomatal parameters applied to all biomes over the world. In fact, the large model-observation mismatch over ATTO (fig. 5), which the authors attribute to underestimated stomatal uptake (line 327), may also be a product improper parameterization more than inaccurate meteorology.

The same problem happens similarly, but to a lesser extent, for the cuticular parameterization, as Zhang et al. (2003) did assign different cuticular uptake parameters for different land types. But it is much more difficult to assess whether these parameters make sense than their stomatal counter parts. So this should be a minor issue. However, some discussions on the uncertainty and inter-biome variability of these parameters is important.

Another main issue is the model evaluation, which may also stem from the fact that the proposed scheme has no biome dependence. The model evaluation over the four sites is mostly specific and well-thought.

However, in most recent work involves evaluating (Silva and Heald, 2018; Wong et al., 2019), developing (Clifton et al., 2020b; Lin et al., 2019) or reviewing (Clifton et al., 2020a) dry deposition schemes, extensive effort have been done to compile worldwide ozone dry deposition measurements to gauge the performance of ozone over different biomes. Most of the above works have publicized their compiled ozone deposition measurements. Adding another part of evaluation that focus on the performance over different land types is necessary in both establishing the credibility of the proposed scheme and identifying its potential weakness, especially given this is a global model.

As both the vertical transfer and canopy resistance schemes are modified, the update should affect not only $O_3$, but all trace gases. It would be interesting to include a brief description on the changes in some other important trace gases (e.g. $NO_2$, $SO_2$, $HNO_3$)

**Technical comments:**

Line 106:

Let's refer to fig. 4 of Baldocchi et al. (1987). Linear scaling always produces lower resistance, and therefore higher uptake, than proper canopy scaling. Therefore linear scaling should overestimate uptake instead of underestimate.

Line 110:

More discussions and acknowledgements on proposed (e.g. Mészáros et al., 2009; Stella et al., 2019) and implemented (e.g. Clifton et al., 2020b) soil deposition schemes are need.

Line 192

How is wetness and snow-covered fraction calculated? How is it related to LAI? These should be clarified.

Line 235

There are also other important long-term measurements (e.g. Blodgett Forest, Harvard Forest). Why do you choose these particular four data sets out of all available ozone flux measurements for detailed evaluation? Additional justification is needed.

Line 254

Non-stomatal deposition does not only include cuticular, but also soil uptake. Other terminology (e.g. total cuticular conductance) shall be used in placed of non-stomatal conductance to avoid confusion and imprecision.

**References**

Baldocchi, D. D., Hicks, B. B. and Camara, P.: A canopy stomatal resistance model for gaseous deposition to vegetated surfaces, Atmos. Environ., 21(1), 91–101, doi:10.1016/0004-6981(87)90274-5, 1987.

Clifton, O. E., Fiore, A. M., Massman, W. J., Baublitz, C. B., Coyle, M., Emberson, L., Fares, S., Farmer, D. K., Gentine, P., Gerosa, G., Guenther, A. B., Helmig, D., Lombardozzi, D. L., Munger, J. W., Patton, E. G., Pusede, S. E., Schwede, D. B., Silva, S. J., Sörgel, M., Steiner, A. L. and Tai, A. P. K.: Dry Deposition of Ozone over Land: Processes, Measurement, and Modeling, Rev. Geophys., 58(1), doi:10.1029/2019rg000670, 2020a.

Clifton, O. E., Paulot, F., Fiore, A. M., Horowitz, L. W., Correa, G., Baublitz, C. B., Fares, S., Goded, I., Goldstein, A. H., Gruening, C., Hogg, A. J., Loubet, B., Mammarella, I., Munger, J. W., Neil, L., Stella, P., Uddling, J., Vesala, T. and Weng, E.: Influence of dynamic ozone dry deposition on ozone pollution, J. Geophys. Res. Atmos., 125(8), e2020JD032398, doi:10.1029/2020jd032398, 2020b.

Emberson, L. D., Kitwiroon, N., Beevers, S., Büker, P. and Cinderby, S.: Scorched earth: How will changes in the strength of the vegetation sink to ozone deposition affect human health and ecosystems?, Atmos. Chem. Phys., 13(14), 6741–6755, doi:10.5194/acp-13-6741-2013, 2013.

Fares, S., Matteucci, G., Scarascia Mugnozza, G., Morani, A., Calfapietra, C., Salvatori, E., Fusaro, L., Manes, F. and Loreto, F.: Testing of models of stomatal ozone fluxes with field measurements in a mixed Mediterranean forest, Atmos. Environ., doi:10.1016/j.atmosenv.2012.11.007, 2013.

Hardacre, C., Wild, O. and Emberson, L.: An evaluation of ozone dry deposition in global scale chemistry climate models, Atmos. Chem. Phys., 15(11), 6419–6436, doi:10.5194/acp-15-6419-2015, 2015.

Hoshika, Y., Osada, Y., de Marco, A., Peñuelas, J. and Paoletti, E.: Global diurnal and nocturnal parameters of stomatal conductance in woody plants and major crops, Glob. Ecol. Biogeogr., 27(2), 257–275, doi:10.1111/geb.12681, 2018.

Jarvis, P. G.: The Interpretation of the Variations in Leaf Water Potential and Stomatal Conductance Found in Canopies in the Field, Philos. Trans. R. Soc. B Biol. Sci., 273(927), 593–610, doi:10.1098/rstb.1976.0035, 1976.

Lin, M., Malyshev, S., Shevliakova, E., Paulot, F., Horowitz, L. W., Fares, S., Mikkelsen, T. N. and Zhang, L.: Sensitivity of ozone dry deposition to ecosystem-atmosphere interactions: A critical appraisal of observations and simulations, Global Biogeochem. Cycles, 33, doi:10.1029/2018gb006157, 2019.

Mészáros, R., Horváth, L., Weidinger, T., Neftel, A., Nemitz, E., Dämmgen, U., Cellier, P. and Loubet, B.: Measurement and modelling ozone fluxes over a cut and fertilized grassland, Biogeosciences, 6(10), 1987–1999, doi:10.5194/bg-6-1987-2009, 2009.

Silva, S. J. and Heald, C. L.: Investigating Dry Deposition of Ozone to Vegetation, J. Geophys. Res. Atmos., 123(1), 559–573, doi:10.1002/2017JD027278, 2018.

Simpson, D., Benedictow, A., Berge, H., Bergström, R., Emberson, L. D., Fagerli, H., Flechard, C. R., Hayman, G. D., Gauss, M., Jonson, J. E., Jenkin, M. E., Nyúri, A., Richter, C., Semeena, V. S., Tsyro, S., Tuovinen, J. P., Valdebenito, A. and Wind, P.: The EMEP MSC-W chemical transport model – Technical description, Atmos. Chem. Phys., 12(16), 7825–7865, doi:10.5194/acp-12-7825-2012, 2012.

Stella, P., Loubet, B., de Berranger, C., Charrier, X., Ceschia, E., Gerosa, G., Finco, A., Lamaud, E., Serça, D., George, C. and Ciuraru, R.: Soil ozone deposition: Dependence of soil resistance to soil texture, Atmos. Environ., doi:10.1016/j.atmosenv.2018.11.036, 2019.

Wang, Y., Jacob, D. J. and Logan, J. A.: Global simulation of tropospheric O 3-NO x-hydrocarbon chemistry - 1. Model formulation, J. Geophys. Res. D Atmos., 103(3339), 10713–10725, doi:10.1029/98jd00158, 1998.

Wesely, M. L.: Parameterization of surface resistances to gaseous dry deposition in regional-scale numerical models, Atmos. Environ., 41(SUPPL.), 52–63, doi:10.1016/j.atmosenv.2007.10.058, 1989.

Wong, A. Y. H., Geddes, J. A., Tai, A. P. K. and Silva, S. J.: Importance of dry deposition parameterization choice in global simulations of surface ozone, Atmos. Chem. Phys., 19(22), 14365–14385, doi:10.5194/acp-19-14365-2019, 2019.

Zhang, L., Brook, J. R. and Vet, R.: A revised parameterization for gaseous dry deposition in air-quality

models, Atmos. Chem. Phys., 3(6), 2067–2082, doi:10.5194/acp-3-2067-2003, 2003.

---

## Referee Comment (RC3) · Anonymous Referee #3 · 5 Oct 2020

Comments on:

**A revised dry deposition scheme for land-atmosphere exchange of trace gases in ECHAM/MESSy v2.54**

Emmerichs et al.,

submitted to Geophysical Model Development, June 2020

General comments:

This manuscript presents a revised dry deposition scheme in EMAC model. Revision includes improved stomatal deposition dependence on vegetation density, two meteorological factors and soil moisture, and improved non-stomatal deposition dependence on meteorological and environmental factors. Then authors evaluated the impact of this revision on ozone dry deposition on daily and seasonal scales and explored the effect of each separate parameterization.

In general, I think this manuscript is really well written, and it meets the criteria for publication on Geophysical Model Development:

- the manuscript contributes to the modeling of boundary layer dynamics and atmospheric chemistry
- scientific approach and methods used are valid, results are discussed in an appropriate way
- simulation is reproducible because of data and code availability, and detailed description in the main manuscript and supplements

Just a few questions and details need to be further addressed before accepting for publication:

**Specific comments**:

- Abstract could be revised to better summarize (shorten) the model development details and leave room for model performance improvement and impact on global ozone budget, etc.
- Authors selected 6 land types out of 11 in the model. Could authors add in the reason for including or excluding certain land types?
- Mismatching meteorology: in Sect. 4.2, authors choose/have to use meteorology data from ERA5 to assess the impact of stress factors on the diurnal cycle of dry

deposition. And in line 371-372, '…, as the humidity over the Amazon forest **is probably too low** in the model'. Same argument is presented in line 415-417. Could these mismatches/comparisons in meteorology be shown in appendix as figures?

**Technical corrections**:

- Line 92, 'The the'.
- Line 286, is 'what' a typo here?

---

## Author Comment (AC2) · 23 Oct 2020

**Answer to referee comment 3**

October 23, 2020

Dear referee, many thanks for the comments. We appreciate that the referee recognise the purpose and the importance of this manuscript. Please find our replies to the special comments below.

> *Special comments*
> *Abstract could be revised to better summarize (shorten) the model development details and leave room for model performance improvement and impact on global ozone budget, etc.*

Reply: We agree that the abstract can be more concise to include additional important aspects you mentioned. We will re-write it.

> *Authors selected 6 land types out of 11 in the model. Could authors add in the reason for including or excluding certain land types?*

Reply: We used the already existing surface scheme in MESSy for this study. The 6 land types (better termed surface types) is a generalisation of the originally given 11 types ( (1) Urban land, (2) agricultural land, (3) range land, (4) deciduous forest, (5) coniferous forest, (6) mixed forest including wetland, (7) water including both salt and fresh, (8) barren land - mostly desert, (9) non-forested wetland, (10) mixed agricultural and range land, (11) rocky open areas with low-growing shrubs) whereas e.g. the here used surface type vegetation represents all vegetated areas.

> *Mismatching meteorology: in Sect. 4.2, authors choose/have to use meteorology data from ERA5 to assess the impact of stress factors on the diurnal cycle of dry deposition. And in line 371-372, '..., as the humidity over the Amazon forest is probably too low in the model'. Same argument is presented in line 415-417. Could these mismatches/comparisons in meteorology be shown in appendix as figures?*

Reply: Yes, the mentioned aspect plays an important role for the analysis at ATTO. The argumentation can be illustrated and clarified with figures of the comparison of the meteorology. We will add these figures to the appendix.

The typos mentioned in the technical comments will be corrected.

---

## Author Comment (AC3) · 23 Oct 2020

**Answer to referee comment 2**

October 23, 2020

Dear referee,
we are thankful for the detailed review. Our replies are below:

> **General Comments:**
> *Even there is always a severe lack of direct flux measurement, the sporadic efforts over the past 20 years still reveals a lot of new and interesting environment dependence and inter-site variabilities of gaseous dry deposition after the proposal of the ever-popular (Hardacre et al., 2015) Wesely scheme (Wesely, 1989) and its slight variants (e.g. Wang et al., 1998). Meanwhile, enormous advance has been made over modelling carbon-water exchange, and therefore stomatal modelling. And given that dry deposition has been shown as one of the major uncertainty of modelling surface ozone (Wong et al., 2019), therefore, I largely agree with the position of the first reviewer, that the effort of updating gaseous dry deposition schemes shall be welcomed and encouraged.*

Reply: In the context of developing an atmospheric (chemical) model we chose to extend the common Wesely scheme of MESSy with well-known empirical relationships. The extension firstly captures the dependency on vegetation density, heat and drought which have been shown to be major drivers of inter-site variability's (Wong et al., 2019, Hardacre et al., 2015). Modelling the stomatal behaviour with more mechanistic models, e.g. based on carbon assimilation is a subject of future developments in MESSy. A paragraph on these future developments will be added as manuscript outlook.

> *Yet, I doubt whether this paper is doing a good enough job in "updating" the dry deposition scheme, particularly in terms of modelling canopy resistance. Given the functional diversity of plants on the Earth, I find one of the biggest weakness of the scheme presented in this paper is the lack of biome-dependence of both its stomatal*

*and cuticular parameters, especially given that previous works have already addressed this issue (e.g. Emberson et al., 2013; Simpson et al., 2012; Zhang et al., 2003). There is also notable weakness in evaluation of the proposed scheme, but it is much easier to address.*

Reply: We understand the reviewer doubts when comparing to the dry deposition scheme of other current models. However, the implementations of stomatal conductance dependence on vegetation density, heat and drought stress as well as cuticular uptake linked to meteorology introduce firstly important functionalities of dry deposition at vegetation to MESSy. Although the scheme is still only based on four different surface types these revision represents a significant advancement for dry deposition modelling with MESSy allowing a more realistic account of an important global ozone sink. Thereby, MESSy still lacks a detailed and mechanistic description of terrestrial vegetation that is evaluated and routinely used by the MESSy community. The documentation, evaluation and publication of the developments presented in the manuscript are important beyond the MESSy community. In fact EMAC participates to the world wide Model Intercomparison Projects (not at least CMIP6), where the full documentation of the models published is essential to understand differences among the different models. To provide a platform for this kind of model description is one of the goals of GMD. Implementing a biome-dependent dry deposition model coupled to $CO_2$ assimilation (White et al. 2004) is planned as a follow-up development in MESSy. Biome-dependent vegetation cover information, required for this scheme, are then provided by global input data which, however, represent only the annual cycle of vegetation. The recently available dynamic vegetation model LPJ-GUESS providing detailed vegetation information with the temporal variability required for a climate model could be a further improvement. By now the one-way coupling of LPJ-GUESS as a MESSy submodel is only in the initial evaluation of the coupling with the atmospheric model (Forrest et al., 2020). A description of these future developments will be added as an outlook section to the manuscript.

> **Specific Comments:**
> *Starting from stomatal conductance. I agree with the authors, that the simplicity and effectiveness of Jarvis-type parameterizations have its place in atmospheric modelling. Yet this particular ecophysiological theory itself (Jarvis, 1976) only states that stomatal conductance has multiple simultaneous constraint (mathematically, $g_s = g_{max} \prod_{i}^{n_{constraints}} f_i(X_i), 0 \leq f_i \geq 1$) but does not explicitly gives universal functional (i.e. the mathematical forms of $f_i$) and parameters of all biomes over the world. It has been explicitly shown that improperly parameterized Jarvis-type model can lead to substantial bias (Fares et al., 2013).*

Reply: We are aware of the limitations of the Jarvis-type model but among others Fares et al. (2013) showed that the Jarvis-type model captured measured O3 dry deposition fluxes better than a Ball-Berry model based on CO2 assimilation. The criticism of the Jarvis-type model in Fares et al. (2013) concerns the missing ability of the VPD factor in representing the 'VPD driven afternoon depression'. However, we used instead of the proposed drought stress factor by Jarvis the mechanistic factor based on the optimised exchange of CO2 and water by plants (Katul et al. 2009). We will add a section on the uncertainties and limitations of the Jarvis-type model to the manuscript.

> *Earlier works of updating dry deposition schemes with Jarvis-type stomatal sub-models (e.g. Simpson et al., 2012; Zhang et al., 2003) had already been assigning stomatal parameters to each individual biome. Though one may argue that they are neither backed empirically (improperly parameterized), they are probably still working better, especially for global modelling, than one single set of stomatal parameters over all biomes. For example, Hoshika et al.(2018) empirically derive that $g_{smax}$ (maximum stomatal conductance) can vary almost ten-folds across all biomes, and the optimal temperature of stomatal opening ($T_{opt}$) generally increases as the mean annual air temperature. The Zhang and EMEP parameterizations stated above a re-able to qualitatively capture some features showing in Hoshika et al. (2018) (e.g. higher $g_{smax}$ for broadleaf trees and crops than boreal forests, higher $T_{opt}$ for tropical than boreal biomes), giving them more creditability when applied regionally and globally, which cannot be achieved by one single set of stomatal parameters applied to all biomes over the world.*

Reply: We see the importance of the biome-dependent parameters which however can introduce uncertainties since they are assigned to measurements whereas the absolute values are influenced by multiple factors like genotype and local climatic conditions (Sulis et at., 2015; Hoshika et al., 2018, Tuovinen et al., 2009). Admittedly, detailed parameters are presented in e.g. LRTAP (2009) but for large-scale models with their limitations they have to be simplified like it is done for the EMEP model (Simpson et al. 2012). The most sensitive and uncertain parameter for dry deposition modelling at stomata $g_{smax}$ is not used. Instead, we parametrized the background stomatal behaviour explicitly depending on the photosynthetically active radiation according to Sellers (1985). Regarding the optimal temperature of stomatal behaviour we have to consider that for the maximum and minimum temperature, which are directly related to the optimal temperature, only less measurements under field conditions are available (Hoshika et al., 2018). For these reasons among others, we decided to keep the four-type surface scheme of MESSy for dry deposition modelling in which then biome-dependent parameter sets are not included.

> *In fact, the large model-observation mismatch over ATTO (Fig.*

*5), which the authors attribute to underestimated stomatal uptake (line327), may also be a product improper parameterization more than inaccurate meteorology.*

Reply: Yes, the discrepancy at ATTO could be due to an improper parametrization of stomatal conductance whereas the neglected chemical within-canopy reactions, however, are also an uncertainty source (Freire et al. 2017). On the other hand the biased meteorology and moisture cycling is a well-known issue in ECHAM (Hagemann and Stacke 2015) and plays a role for dry deposition modelling here as well. In Fig. 5b of the manuscript we can show that modified meteorology and transpiration at least partly improves the modelled dry deposition velocity in the Amazon forest.

*The same problem happens similarly, but to a lesser extent, for the cuticular parameterization, as Zhang et al.(2003) did assign different cuticular uptake parameters for different land types. But it is much more difficult to assess whether these parameters make sense than their stomatal counter parts. So this should be a minor issue. However, some discussions on the uncertainty and inter-biome variability of these parameters is important.*

Reply: The cuticular parametrization by Zhang et al. (2002) was implemented in order to account for the second important ozone deposition pathway in our model. This pathway was effectively neglected in the previous model version. As well as for the stomatal uptake we built up on the existing resistance scheme in MESSy which distinguish between only four different surface types. Here we also used less generalised parameters. An overall consideration of the uncertainty and limitations of the used model, however, is important and will be added as a separate section to the manuscript.

*Another main issue is the model evaluation, which may also stem from the fact that the proposed scheme has no biome dependence. The model evaluation over the four sites is mostly specific and well-thought. However, in most recent work involves evaluating (Silva and Heald, 2018; Wong et al., 2019), developing (Clifton et al., 2020b; Lin et al., 2019) or reviewing (Clifton et al., 2020a) dry deposition schemes, extensive effort have been done to compile world-wide ozone dry deposition measurements to gauge the performance of ozone over different biomes. Most of the above works have publicised their compiled ozone deposition measurements. Adding another part of evaluation that focus on the performance over different land types is necessary in both establishing the credibility of the proposed scheme and identifying its potential weakness, especially given this is a global model.*

Reply: The whole data comparison at the four chosen sites account for the most important high vegetation covered biomes on the Earth. For the reason of uniqueness and importance to investigate atmospheric processes in a remote and pristine forest the Amazonian Tall Tower Observatory (ATTO) stands out. In order to include an analysis at this site in our study we adapt the choice of the simulation period to the availability of measurements there, specifically. The used and described measurement data listed in e.g. Clifton et al. (2020a) have been obtained in the late 2000s and early 2010s. However, the analysis period should cover the recent decade which includes most extreme drought and heat events (where the stomatal stress factors are aimed for). Moreover, since we consider the inter-annual differences at the different locations we only compare data which cover the same time period. Including further measurement sites would require a new simulations.

> *As both the vertical transfer and canopy resistance schemes are modified, the update should affect not only O3, but all trace gases. It would be interesting to include a brief description on the changes in some other important trace gases (e.g. $NO_2$, $SO_2$, $HNO_3$).*

Reply: The changes, indeed affect trace gases other than ozone. However, this manuscript focuses on ozone because among it's atmospheric importance the applied Wesely scheme is based on the the dry deposition mechanism of ozone (Wesely 1989). By including the changes in $O_x$ budget, that includes $NO_2$ and $HNO_3$, we cover many important tropospheric trace gases. We will further add a figure with the changes for the fluxes of $NO_2$, $HNO_3$, HCHO and $SO_2$ and the respective description.

> ### Technical comments
> *Line 106:*
> *Let's refer to Fig. 4 of Baldocchi et al.(1987). Linear scaling always produces lower resistance, and therefore higher uptake, than proper canopy scaling. Therefore linear scaling should overestimate uptake instead of underestimate.*

Reply: Indeed, the linear scaling lead to an overestimation of the uptake. Thank you for pointing to this typo.

> *Line 110:*
> *More discussions and acknowledgements on proposed (e.g. Mészáros et al., 2009; Stella et al., 2019) and implemented (e.g. Clifton et al., 2020b) soil deposition schemes are need.*

Reply: We can add discussions and acknowledgements on existing soil deposition parametrizations.

*Line 192:*
*How is wetness and snow-covered fraction calculated? How is it related to LAI? These should be clarified.*

Reply: The wet skin fraction is calculated from the wet skin reservoir ($wl$ [m]) and Leaf Area Index ($LAI$ [$m^2/m^2$]):

$$cvw \sim wl/(1 + LAI)$$

whereas the snow covered fraction depends mainly on the snow at the surface ($h_s$ [m water equivalent]):

$$cvs \sim \tanh{(h_s)}\sqrt{h_s}$$

The detailed description can be found in the documentation of ECHAM5 (Klimarechenzentrum 1992 eq. 3.3.2.4; Roeckner et al., 2003 eq. 6.45 )

*Line 235:*
*There are also other important long-term measurements (e.g. Blodgett Forest, Harvard Forest). Why do you choose these particular four data sets out of all available ozone flux measurements for detailed evaluation? Additional justification is needed.*

Reply: We reviewed and ask for several data sets. The chosen data sets were the best available of ozone dry deposition (flux data and ozone mixing ratio or velocity data) with the required temporal resolution and coverage which also represent different parts and biomes of the world. As examples, for Harvard forest data of O3 dry deposition flux and O3 mixing ratio is only available until 1997[1]) whereas at Blodgett forest the total measuring period (2001-2007) doesn't match the chosen simulation period. Like described above we didn't use data with non-matching time coverage since we consider inter-annual differences at the measurement sides.

*Line 254:*
*Non-stomatal deposition does not only include cuticular, but also soil uptake. Other terminology (e.g. total cuticular conductance) shall be used in placed of non-stomatal conductance to avoid confusion and imprecision.*

Reply: Yes, at the points where the uptake to the leaf surfaces is meant the term cuticular conductance should be used. However, some cited studies report measurements (partioning) of non-stomatal dry deposition which captures among others the removal at the cuticle. We will clearly distinguish this terms.
* * *
[1]data coverage of 'O3.mlb' and f.o3 is shown in plot 3 and 5 in https://harvardforest1.fas.harvard.edu/sites/harvardforest.fas.harvard.edu/files/data/plots/hf004-01.pdf

**References**

Forrest, Matthew, et al. "Including vegetation dynamics in an atmospheric chemistry-enabled general circulation model: linking LPJ-GUESS (v4. 0) with the EMAC modelling system (v2. 53)." Geoscientific Model Development 13.3 (2020).

Freire, L. S., et al. "Turbulent mixing and removal of ozone within an Amazon rainforest canopy." Journal of Geophysical Research: Atmospheres 122.5 (2017): 2791-2811.

Hagemann, Stefan, and Tobias Stacke. "Impact of the soil hydrology scheme on simulated soil moisture memory." Climate Dynamics 44.7-8 (2015): 1731-1750.

Hardacre, C., O. Wild, and L. Emberson. "An evaluation of ozone dry deposition in global scale chemistry climate models." Atmospheric Chemistry and Physics 15.11 (2015): 6419-6436.

LRTAP: "Mapping critical levels for vegetation, in: Manual on Methodologies and Criteria for Mapping CriticalLoads and Levels and Air Pollution Effects, Risks and Trends." Revision of 2009, edited by Mills, G., UNECE Convention on Long-range Transboundary Air Pollution. International Cooperative Programme on Effects of AirPollution on Natural Vegetation and Crops

Klimarechenzentrum, Deutsches. "The ECHAM3 atmospheric general circulation model." Techn. Rep 6 (1992).

Hoshika, Yasutomo, et al. "Global diurnal and nocturnal parameters of stomatal conductance in woody plants and major crops." Global ecology and biogeography 27.2 (2018): 257-275.

Roeckner, Erich, et al. "The atmospheric general circulation model ECHAM 5. PART I: Model description." (2003).

Sellers, Piers J. "Canopy reflectance, photosynthesis and transpiration." International journal of remote sensing 6.8 (1985): 1335-1372. Wong, Anthony Y., et al. "Importance of dry deposition parameterization choice in global simulations of surface ozone." Atmospheric Chemistry and Physics 19.22 (2019).

Sulis, M.; Langensiepen, M.; Shrestha, P.; Schickling, A.; Simmer, C. Kollet, S. J. Evaluating the Influence of Plant-Specific Physiological Parameterizations on the Partitioning of Land Surface Energy Fluxes J. Hydrometeor, American Meteorological Society, 2015, 16, 517-533

Tuovinen, Juha-Pekka, Lisa Emberson, and David Simpson. "Modelling

ozone fluxes to forests for risk assessment: status and prospects." Annals of Forest Science 66.4 (2009): 1-14.

Wesely, Ml. "Parameterization of surface resistances to gaseous dry deposition in regional-scale numerical models." Atmospheric environment 23.6 (1989): 1293-1304.

Zhang, Leiming, Jeffrey R. Brook, and Robert Vet. "On ozone dry deposition—with emphasis on non-stomatal uptake and wet canopies." Atmospheric Environment 36.30 (2002): 4787-4799.
White, P. W. "IFS documentation CY23r4: Part IV physical processes." (2004).

---

## Author Response (AR1)

**1 Comment 1 by Dennis Baldocchi**

**1.1 Comment from Referee**

The field of dry deposition has had periods of ups and downs in activity and research. Unfortunately algorithms in important models have been fossilized to consider the Wesely model of 1989. While that was a very good and appropriate algorithm 30 years ago, we know more about land surface fluxes, how to model stomatal conductance and have been datasets and parameterization information in 2020. So, I was excited to see this paper.

I see the main contributions are

The default dry deposition scheme has been extended with adjustment factors to predict stomatal responses to temperature and vapour pressure deficit. Furthermore, an explicit formulation of the non-stomatal deposition to the leaf surface (cuticle) dependent on humidity has been implemented based on established schemes. Finally, the soil moisture availability function for plants has been revised to be consistent with the simple hydrological model available in EMAC.

The authors make a good case for this work and its significance as 'the revision of the process parameterisation as documented here has the potential to significantly reduce the overestimation of tropospheric ozone in global models'.

This paper is a steps in the right direction, but revolves around the over parameterized Jarvis stomatal model that was used in the 80s with more adjustment factors. Many of us, including Piers Sellers, have abandoned the Jarvis model in land-surface modeling of water and carbon fluxes because it lead to stomatal suicide. Others have adopted the Ball-Berry approach, with better fidelity

Baldocchi, D. D., and T. Meyers (1998), On using eco-physiological, micrometeorological and biogeochemical theory to evaluate carbon dioxide, water vapor and trace gas fluxes over vegetation: a perspective, Agricultural and Forest Meteorology, 90(1-2), 1-25.

I don't view this 'new'model as an improvement by going back to the Jarvis model for stomatal conductance. There has been many advances in stomatal modeling worth considering in 2020.

Wang, Yujie, John S. Sperry, William RL Anderegg, Martin D. Venturas, and Anna T. Trugman. "A theoretical and empirical assessment of stomatal optimization modeling." New Phytologist (2020).

Medlyn, B. E., Duursma, R. A., Eamus, D., Ellsworth, D. S., Prentice, I. C., Barton, C. V., ... & Wingate, L. (2011). Reconciling the optimal and empirical approaches to modelling stomatal conductance. Global Change Biology, 17(6), 2134-2144.

Personally, I'd like to see some connection with ecosystem photosynthesis scaling with stomatal conductance. There has been excellent advances modeling both that could be coupled with a stomatal and dry deposition model, for instance.

Jiang, C., and Y. Ryu (2016), Multi-scale evaluation of global gross primary productivity and evapotranspiration products derived from Breathing Earth System Simulator (BESS), Remote Sensing of Environment, 186, 528-547, doi:http://dx.doi.org/10.1016/j.rse.2016.08.030.

De Kauwe, Martin G., et al. "A test of an optimal stomatal conductance scheme within the CABLE land surface model." Geoscientific Model Development (2015): 431-452.

In writing the introduction, there has been some recent workshops on dry deposition, newer long term studies and a very good review that should be cited and considered

Clifton, O. E., Fiore, A. M., Massman, W. J., Baublitz, C. B., Coyle, M., Emberson, L., ... (2020). Dry deposition of ozone over land: processes, measurement, and modeling. Reviews of Geophysics, 58(1), e2019RG000670.

Clifton, O. E., A. M. Fiore, J. W. Munger, S. Malyshev, L. W. Horowitz, E. Shevliakova, F. Paulot, L. T. Murray, and K. L. Griffin (2017), Interannual variability in ozone removal by a temperate deciduous forest, Geophysical Research Letters, 44(1), 542-552, doi:10.1002/2016gl070923.

Clifton, O. E., Paulot, F., Fiore, A. M., Horowitz, L. W., Correa, G., Baublitz, C. B., ... (2020). Influence of dynamic ozone dry deposition on ozone pollution. Journal of Geophysical Research: Atmospheres, 125(8), e2020JD032398.

I am of mixed feelings of this work. I find the model algorithm dated and not an improvement. On the other hand there has been a dearth of long term flux measurements and use of those data to test the performance of a model, as it done here.
To my opinion this would be much better paper by using modern, better state of art stomatal models that couple carbon and water fluxes and test the performance against a year of flux measurements. Then I would feel the work is new, novel and a significant improvement over the past work.
I also like the use of 4 contrasting flux datasets. This too is an advance in model testing. For example regarding performance, we learn 'As seen from the comparison of stomatal resistance values (Fig. 4d) the model underestimates the stomatal uptake. This is because the irrigation of the Orchard leads to cooling sustained evapotranspiration and keeps $f(T)$ low.Thus in the model, a too high temperature stress act on the stomata'.
My alternative hypothesis is that this bias may disappear with a coupled carbon-water stomatal conductance model.
If I have learned anything over my career it is the power and importance of multiple constraints. Sadly, the Jarvis model does not deliver. It was great circa 1976 and helped us think about the role of stomata on dry deposition in the 1980s, but that is its extent of being good enough.

Fig 3 would be better if error bars were added, given these are monthly means. I do like the global upscaling. It helps address the 'so what?' question and does produce some multiple constraint with regards to getting pollution right, as we see in Fig. 6.

My bottom line is that this paper can be remedied. It has lots of strengths worth keeping. And the spirit of the work is good.

Regarding conclusion

The seasonal variability of the simulated dry deposition velocity could be further improved by using as model input the time-series of vegetation cover from an imaging products which also capture land use changes and vegetation trend that are known to impact dry deposition significantly.

Connection to phenology modeling or observation is key to getting the seasonality in LAI correct and the fluxes right. So Yes this is an important aspect of the model. I'd like to see it in the 'new model'.

If the model had already coupled water and carbon phenology should be part of it.

**1.2   Author's response**

Dear referee, many thanks for your review. Here are our replies: The article documents a revision of the existing dry deposition scheme in EMAC not a complete new implementation. The idea is to improve the existing scheme based on the already available information in the model (i.e. without detailed phenology information etc.) because model results show that a more precise representation could lower the overestimation of ozone by models. With the current model version, these developments can only draw on limited vegetation information without details on cover and phenology. Dry deposition of trace gases is represented by the "resistance-in-series" scheme of Wesely (1989). The stomatal uptake was firstly only based on the response to incoming solar radiation developed by Sellers (1985) which is known to be an important fluctuation factor (Dawson et al. 2010), and a soil moisture stress factor. The further developments were build on this common dry deposition scheme. For the extension with additional stress factors, we adopt the multiplicative principle and the temperature stress factor by Jarvis (1976). This principle is commonly used in second-generation LSM schemes due to its computational efficiency, adaptability and simplicity (Pitman et al. 2003, Clifton et al. 2020) and has been shown to capture 95 % of the observed variability of stomatal conductance (Dawson et al 2010). The stomatal sensitivity to vapour pressure deficit is calculated according to the optimisation framework by Katul et al. (2009) which maximises the use of carbon under a minimal cost of water inside the plant. This concept accounts for the water cost of carbon without specifying the stomatal response to VPD and $CO_2$ in advance and agrees well with experimental data (Katul et al. 2009). Hence, by adding also the stomatal response to temperature and vapour pressure deficit within this study, the key responses of stomates are represented (Pitman et al. 2003). Comparing to measurement data, several studies found that Jarvis-type models can compete with Ball-Berry models in explaining observed stomatal

conductance and stomatal ozone flux to vegetation (Hoshika et al. 2017, Ran et al. 2017) whereas both have different limitations and advantages (Lu 2018, Farquhar et al. 1980). The performance of both models depend certainly on the choice of parameters (Sulis et al. 2015, Lu 2018). The mentioned "stomatal suicide" as major critique to the Jarvis model has been experienced in EMAC and is attributed to the lack of soil moisture storage in some regions. It is solved currently by adapting the soil moisture stress factor to the used soil representation. Moreover, the stress factor dependent on VPD (Katul et al. 2009), that we use, exerts a stronger control on evapostranspiration that the original factor proposed by Jarvis. For comparison, at VPD = 5 kPa stomatal conductance is predicted to decrease by about 50% and < 10% according to Katul et al. (2009) and Jarvis (1976), respectvely. A further amelioration of the EMAC model dry bias in the Amazon is brought by the use of VPD factor by Katul et al. (2009) only in simulations without meteorological nudging (not shown in the manuscript). The usage of the Ball-Berry approach is constrained by the availability of detailed information on plant microphysics which determine the parameters. Due to the current limitations of EMAC in this regard, described above, an implementation would build on many assumptions concerning the representation at global scale.

With regard to the developments of stomatal conductance models in the last years the approach used here is dated but in EMAC this represents a significant improvement compared to the existing parametrization.The adaptability, simplicity and computer efficiency makes it attractive for the use at global scale and the usage of parametrizations for radiation response and VPD stress are different from the one used in Jarvis (1976).

We agree with the Referee but unfortunately these developments are limited by the minimal ecosystem representation in the EMAC model. Implementing a mechanistic approach which connect stomatal conductance to plant photosynthesis is definitely intended for EMAC once a vegetation model with the sufficient details and well-constrained parameters will be available.

We will add a paragraph on the current research status of dry deposition to the introduction considering this studies .

With regard to the mentioned limitations and the current status of the dry deposition parametrization in EMAC, our development can be seen as an intermediate stage on the way to a "state-of-the-art" dry deposition scheme. For the stomatal part, major dependencies to meteorology have been established whereas the implementation of the cuticular pathway contributes to a global enhancement of dry deposition especially of soluble organic species that are ozone precursors. Furthermore, the study has a significance for the MESSy community as first technical description and evaluation of the vertical exchange submodel VERTEX.

Concerning the model evaluation at Citrus Orchard, we cannot exclude that such a model might remove the bias. However, if it did, it would do it for the wrong reasons. The absence of soil water stress at Citrus Orchard (due to irrigation) is artificial and not represented in the global model. Thus, the site cannot be representative for the mostly non irrigated 1.1°x1.1° grid box

including Citrus Orchard. In fact, removal of the water stress from the model greatly reduces the model bias at Citrus Orchard (see Fig. 4d).

We are aware of the limitations of the implemented model parametrization. But regarding that the developments for a global model which has only a minimal ecosystem representation available, we see the current implementation as the best achievable in EMAC without having to embark on the coupling with a dynamic vegetation model that would provide the desired constraints.

Error bars can be added for all sub figures.

Thank you for mentioning this aspect which addresses the actual motivation of this model study. EMAC is an Atmospheric Chemistry Model which explicit chemistry and misses on the other hand details for e.g. the vegetation representation. Regarding all the arguments mentioned above we can not be sure that implementing a simple 'Anet-$g_s$' stomatal approach relying on the scanty vegetation information available in the model could improve the representation of dry deposition in EMAC.

The usage of the time-series of vegetation cover from the Moderate Resolution Imaging Spectroradiometer (MODIS) is in preparation as one of the few available means to represent ecosystem phenology in the current model. However, so far only LAI data from MODIS is available in the model and remaining data like canopy height still have to be acquired. Water and carbon phenology is unfortunately not yet part of the model and will be added as part of a future planned vegetation model for EMAC.

- Section 6 'Uncertainties in modelling stomatal conductance' was added.
- Error bars were added to Figure 3c-f

**2 Comment 2**

**2.1 Comment from Referee**

**General Comments**

Even there is always a severe lack of direct flux measurement, the sporadic efforts over the past 20 years still reveals a lot of new and interesting environment dependence and inter-site variabilities of gaseous dry deposition after the proposal of the ever-popular (Hardacre et al., 2015) Wesely scheme (Wesely, 1989) and its slight variants (e.g. Wang et al., 1998). Meanwhile, enormous advance has been made over modelling carbon-water exchange, and therefore stomatal modelling. And given that dry deposition has been shown as one of the major uncertainty of modelling surface ozone (Wong et al., 2019), therefore, I largely agree with the position of the first reviewer, that the effort of updating gaseous dry deposition schemes shall be welcomed and encouraged.

Yet, I doubt whether this paper is doing a good enough job in "updating" the dry deposition scheme, particularly in terms of modelling canopy resistance. Given the functional diversity of plants on the Earth, I find one of the biggest weakness of the scheme presented in this paper is the lack of biome-dependence of both its stomatal and cuticular parameters, especially given that previous works have already addressed this issue (e.g. Emberson et al., 2013; Simpson et al., 2012; Zhang et al., 2003). There is also notable weakness in evaluation of the proposed scheme, but it is much easier to address.

**Specific Comments**

Starting from stomatal conductance. I agree with the authors, that the simplicity and effectiveness of Jarvis-type parameterizations have its place in atmospheric modelling. Yet this particular ecophysiological theory itself (Jarvis, 1976) only states that stomatal conductance has multiple simultaneous constraint (mathematically, $g_s = g_{max} \prod_{i}^{n_{constraints}} f_i(X_i), 0 \leq f_i \geq 1$al functional forms (i.e. the mathematical forms of fi) and parameters of all biomes over the world. It has been explicitly shown that improperly parameterized Jarvis-type model can lead to substantial bias (Fares et al., 2013).

Earlier works of updating dry deposition schemes with Jarvis-type stomatal sub-models (e.g. Simpson et al., 2012; Zhang et al., 2003) had already been assigning stomatal parameters to each individual biome. Though one may argue that they are neither backed empirically (improperly parameterized), they are probably still working better, especially for global modelling, than one single set

of stomatal parameters over all biomes. For example, Hoshika et al. (2018) empirically derive that gsmax (maximum stomatal conductance) can vary almost ten-folds across all biomes, and the optimal temperature of stomatal opening (Topt) generally increases as the mean annual air temperature. The Zhang and EMEP parameterizations stated above are able to qualitatively capture some features showing in Hoshika et al. (2018) (e.g. higher gsmax for broadleaf trees and crops than boreal forests, higher Topt for tropical than boreal biomes), giving them more creditability when applied regionally and globally, which cannot be achieved by one single set of stomatal parameters applied to all biomes over the world. In fact, the large model-observation mismatch over ATTO (fig. 5), which the authors attribute to underestimated stomatal uptake (line 327), may also be a product improper parameterization more than inaccurate meteorology. The same problem happens similarly, but to a lesser extent, for the cuticular parameterization, as Zhang et al. (2003) did assign different cuticular uptake parameters for different land types. But it is much more difficult to assess whether these parameters make sense than their stomatal counter parts. So this should be a minor issue. However, some discussions on the uncertainty and inter-biome variability of these parameters is important.

Another main issue is the model evaluation, which may also stem from the fact that the proposed scheme has no biome dependence. The model evaluation over the four sites is mostly specific and well-thought. However, in most recent work involves evaluating (Silva and Heald, 2018; Wong et al., 2019), developing (Clifton et al., 2020b; Lin et al., 2019) or reviewing (Clifton et al., 2020a) dry deposition schemes, extensive effort have been done to compile worldwide ozone dry deposition measurements to gauge the performance of ozone over different biomes. Most of the above works have publicized their compiled ozone deposition measurements. Adding another part of evaluation that focus on the performance over different land types is necessary in both establishing the credibility of the proposed scheme and identifying its potential weakness, especially given this is a global model.

As both the vertical transfer and canopy resistance schemes are modified, the update should affect not only O3, but all trace gases. It would be interesting to include a brief description on the changes in some other important trace gases (e.g. $NO_2$, $SO_2$, $HNO_3$).

**Technical comments:**
Line 106:
Let's refer to fig. 4 of Baldocchi et al. (1987). Linear scaling always produces lower resistance, and therefore higher uptake, than proper canopy scaling. Therefore linear scaling should overestimate uptake instead of underestimate.
Line 110:
More discussions and acknowledgements on proposed (e.g. Mészáros et al., 2009; Stella et al., 2019) and implemented (e.g. Clifton et al., 2020b) soil deposition schemes are need.
Line 192:
How is wetness and snow-covered fraction calculated? How is it related to LAI?

These should be clarified.
Line 235:
There are also other important long-term measurements (e.g. Blodgett Forest, Harvard Forest). Why do you choose these particular four data sets out of all available ozone flux measurements for detailed evaluation? Additional justification is needed.
Line 254:
Non-stomatal deposition does not only include cuticular, but also soil uptake. Other terminology (e.g. total cuticular conductance) shall be used in placed of non-stomatal conductance to avoid confusion and imprecision.

We understand the reviewer doubts when comparing to the dry deposition scheme of other current models. However, the implementations of stomatal conductance dependence on vegetation density, heat and drought stress as well as cuticular uptake linked to meteorology introduce firstly important functionalities of dry deposition at vegetation to MESSy. Although the scheme is still only based on four different surface types these revision represents a significant advancement for dry deposition modelling with MESSy allowing a more realistic account of an important global ozone sink. Thereby, MESSy still lacks a detailed and mechanistic description of terrestrial vegetation that is evaluated and routinely used by the MESSy community. The documentation, evaluation and publication of the developments presented in the manuscript are important beyond the MESSy community. In fact EMAC participates to the world wide Model Intercomparison Projects (not at least CMIP6), where the full documentation of the models published is essential to understand differences among the different models. To provide a platform for this kind of model description is one of the goals of GMD. Implementing a biome-dependent dry deposition model coupled to CO2 assimilation (White et al. 2004) is planned as a follow-up development in MESSy. Biome-dependent vegetation cover information, required for this scheme, are then provided by global input data which, however, represent only the annual cycle of vegetation. The recently available dynamic vegetation model LPJ-GUESS providing detailed vegetation information with the temporal variability required for a climate model could be a further improvement. By now the one-way coupling of LPJ-GUESS as a MESSy submodel is only in the initial evaluation of the coupling with the atmospheric model (Forrest et al., 2020). A description of these future developments will be added as an outlook section to the manuscript.

We are aware of the limitations of the Jarvis-type model but among others Fares et al. (2013) showed that the Jarvis-type model captured measured O3 dry deposition fluxes better than a Ball-Berry model based on CO2 assimilation. The criticism of the Jarvis-type model in Fares et al. (2013) concerns the missing ability of the VPD factor in representing the 'VPD driven afternoon depression'. However, we used instead of the proposed drought stress factor by Jarvis the mechanistic factor based on the optimised exchange of CO2 and water by plants (Katul et al. 2009). We will add a section on the uncertainties and limitations

of the Jarvis-type model to the manuscript.

We see the importance of the biome-dependent parameters which however can introduce uncertainties since they are assigned to measurements whereas the absolute values are influenced by multiple factors like genotype and local climatic conditions (Sulis et at., 2015; Hoshika et al., 2018, Tuovinen et al., 2009). Admittedly, detailed parameters are presented in e.g. LRTAP (2009) but for large-scale models with their limitations they have to be simplified like it is done for the EMEP model (Simpson et al. 2012). The most sensitive and uncertain parameter for dry deposition modelling at stomata $g_{smax}$ is not used. Instead, we parametrized the background stomatal behaviour explicitly depending on the photosynthetically active radiation according to Sellers (1985). Regarding the optimal temperature of stomatal behaviour we have to consider that for the maximum and minimum temperature, which are directly related to the optimal temperature, only less measurements under field conditions are available (Hoshika et al., 2018). For these reasons among others, we decided to keep the four-type surface scheme of MESSy for dry deposition modelling in which then biome-dependent parameter sets are not included.

Yes, the discrepancy at ATTO could be due to an improper parametrization of stomatal conductance whereas the neglected chemical within-canopy reactions, however, are also an uncertainty source (Freire et al. 2017). On the other hand the biased meteorology and moisture cycling is a well-known issue in ECHAM (Hagemann and Stacke 2015) and plays a role for dry deposition modelling here as well. In Fig. 5b of the manuscript we can show that modified meteorology and transpiration at least partly improves the modelled dry deposition velocity in the Amazon forest.

The cuticular parametrization by Zhang et al. (2002) was implemented in order to account for the second important ozone deposition pathway in our model. This pathway was effectively neglected in the previous model version. As well as for the stomatal uptake we built up on the existing resistance scheme in MESSy which distinguish between only four different surface types. Here we also used less generalised parameters. An overall consideration of the uncertainty and limitations of the used model, however, is important and will be added as a separate section to the manuscript.

The whole data comparison at the four chosen sites account for the most important high vegetation covered biomes on the Earth. For the reason of uniqueness and importance to investigate atmospheric processes in a remote and pristine forest the Amazonian Tall Tower Observatory (ATTO) stands out. In order to include an analysis at this site in our study we adapt the choice of the simulation period to the availability of measurements there, specifically. The used and described measurement data listed in e.g. Clifton et al. (2020a) have been obtained in the late 2000s and early 2010s. However, the analysis period should cover the recent decade which includes most extreme drought and heat events (where the stomatal stress factors are aimed for). Moreover, since we consider the inter-annual differences at the different locations we only compare data which cover the same time period. Including further measurement sites would require a new simulations.

The changes, indeed affect trace gases other than ozone. However, this manuscript focuses on ozone because among it's atmospheric importance the applied Wesely scheme is based on the the dry deposition mechanism of ozone (Wesely 1989). By including the changes in $O_x$ budget, that includes $NO_2$ and $HNO_3$, we cover many important tropospheric trace gases. We will further add a figure with the changes for the fluxes of $NO_2$, $HNO_3$, $HCHO$ and $SO_2$ and the respective description.

Indeed, the linear scaling lead to an overestimation of the uptake. Thank you for pointing to this typo.

We can add discussions and acknowledgements on existing soil deposition parametrizations.

The wet skin fraction is calculated from the wet skin reservoir ($wl$ [m]) and Leaf Area Index ($LAI$ [$m^2/m^2$]:

$$\sim wl/(1 + LAI) \tag{1}$$

whereas the snow covered fraction depends mainly on the snow at the surface ($h_s$ [m water equivalent]):

$$cvs \sim \tanh{(h_s)}\sqrt{h_s} \tag{2}$$

The detailed description can be found in the documentation of ECHAM5 (Klimarechenzentrum 1992 eq. 3.3.2.4; Roeckner et al., 2003 eq. 6.45 )

We reviewed and ask for several data sets. The chosen data sets were the best available of ozone dry deposition (flux data and ozone mixing ratio or velocity data) with the required temporal resolution and coverage which also represent different parts and biomes of the world. As examples, for Harvard forest data of O3 dry deposition flux and O3 mixing ratio is only available until 1997[1]) whereas at Blodgett forest the total measuring period (2001-2007) doesn't match the chosen simulation period. Like described above we didn't use data with non-matching time coverage since we consider inter-annual differences at the measurement sides.

Yes, at the points where the uptake to the leaf surfaces is meant the term cuticular conductance should be used. However, some cited studies report measurements (partioning) of non-stomatal dry deposition which captures among others the removal at the cuticle. We will clearly distinguish this terms.

[revised manuscript text omitted]

**3   Comment 3**

**3.1   Comment from referree**

General comments:
This manuscript presents a revised dry deposition scheme in EMAC model. Revision includes improved stomatal deposition dependence on vegetation density, two meteorological factors and soil moisture, and improved non-stomatal deposition dependence on meteorological and environmental factors. Then authors evaluated the impact of this revision on ozone dry deposition on daily and seasonal scales and explored the effect of each separate parameterization. In general, I think this manuscript is really well written, and it meets the criteria for publication on Geophysical Model Development:
- the manuscript contributes to the modeling of boundary layer dynamics and atmospheric chemistry
- scientific approach and methods used are valid, results are discussed in an appropriate way
- simulation is reproducible because of data and code availability, and detailed description in the main manuscript and supplements

Just a few questions and details need to be further addressed before accepting for publication:

**Specific comments:**

- Abstract could be revised to better summarize (shorten) the model development details and leave room for model performance improvement and impact on global ozone budget, etc.

- Authors selected 6 land types out of 11 in the model. Could authors add in the reason for including or excluding certain land types?

- Mismatching meteorology: in Sect. 4.2, authors choose/have to use meteorology data from ERA5 to assess the impact of stress factors on the diurnal cycle of dry deposition. And in line 371-372, '..., as the humidity over the Amazon forest **is probably too low** in the model'. Same argument is presented in line 415-417. Could these mismatches/comparisons in meteorology be shown in appendix as figures?

**Technical corrections:**

- Line 92, 'The the'.

- Line 286, is 'what' a typo here?

**3.2 Author's response**

Dear referee, many thanks for the comments. We appreciate that the referee recognise the purpose and the importance of this manuscript. Please find our replies to the special comments below. We agree that the abstract can be more concise to include additional important aspects you mentioned. We will re-write it.

We used the already existing surface scheme in MESSy for this study. The 6 land types (better termed surface types) is a generalisation of the originally given 11 types ( (1) Urban land, (2) agricultural land, (3) range land, (4) deciduous forest, (5) coniferous forest, (6) mixed forest including wetland, (7) water including both salt and fresh, (8) barren land - mostly desert, (9) non-forested wetland, (10) mixed agricultural and range land, (11) rocky open areas with low-growing shrubs) whereas e.g. the here used surface type vegetation represents all vegetated areas.

Yes, the mentioned aspect plays an important role for the analysis at ATTO. The argumentation can be illustrated and clarified with figures of the comparison of the meteorology. We will add these figures to the appendix.

The typos mentioned in the technical comments will be corrected.

**3.3 Author's changes in the manuscript**

- In the abstract (Line 9-14):

> .The default dry deposition scheme has been extended with adjustment factors to predict stomatal responses to temperature and vapour pressure deficit. Furthermore, an explicit formulation of the non-stomatal deposition to the leaf surface (cuticle) dependent on humidity has been implemented based on established schemes. Finally, the soil moisture availability function for plants has been revised to be consistent with the simple hydrological model available in EMAC. This revision was necessary in order to avoid unrealistic stomatal closure where the model shows a strong soil dry bias, e.g. in the Amazon basin in the dry season.

were replaced by

> *including meteorological adjustment factors for stomatal closure and an explicit cuticular pathway.*

We added additionally (line 17/18):

> *The scheme is limited by a small number of different surface types and generalised parameters.*

and line 24/25:

> *The change of ozone dry deposition is also reasoned by the altered loss of ozone precursors.*

- In Section 2.2 (line 105/106) we added:

> *This was adapted by Ganzeveld and Lelieveld (1995) to the surface scheme of the ECHAM climate model(Klimarechenzentrum et al., 1992).*

- Doubling 'the' (line 114) was removed
- Line 326: 'what' was changed to 'which'
- Figure A3 on the difference of temperature and relative humidity between EMAC and ERA5 at ATTO was newly included in the appendix (reference in measurement and gl. impact section)

[revised manuscript text omitted]